# DUAL-STAGE FREQUENCY-BASED DENOISING FOR GENERATIVE RECOMMENDATION

## ABSTRACT

Generative recommendation has emerged as a promising frontier in modeling the complex and continuously evolving nature of user preferences. However, its practical effectiveness is often undermined by a fundamental yet overlooked vulnerability: its sensitivity to the pervasive high-frequency sequential noise inherent in raw user interaction data from accidental clicks or transient interests. This paper introduces a paradigm shift that explicitly performs frequency-domain modeling to effectively isolate and suppress sequential noise, while further addressing the challenge of frequency-domain sparsity. Specifically, we propose TONE (Two-stage Optimized deNoising for gEnerative recommendation), a generative framework built around a principled two-stage denoising strategy. In the first stage of item codebook construction, we apply ResGMM (Residual Gaussian Mixture Model) to better fit clustering boundaries, thereby alleviating semantic noise and establishing a robust foundation. In the second stage, on the generative model side, we employ a learnable Gaussian kernel to filter high-frequency noise. Furthermore, we redesign the residual frequency-domain attention mechanism with explicit separation of real and imaginary components, and introduce a learnable matrix to counteract attention collapse induced by Fourier energy concentration, while preserving expressiveness. Empirical results demonstrate that TONE achieves the new state-of-the-art performance over strong baselines on three widely used benchmarks, achieving notable improvements on the Amazon Beauty dataset, with gains of 8.93% in Recall@20 and 8.33% in NDCG@20. Extensive experiments confirm that explicit frequency-domain denoising is key to unlocking a new level of performance and robustness in generative recommendation. The source code is available at `https://anonymous.4open.science/r/TONE-9E07/`.

## 1 INTRODUCTION

In the era of information explosion, recommender systems have become corners of personalized user experiences, seamlessly connecting individuals with relevant content across e-commerce, media, and social platforms(Ko et al., 2022; Wu et al., 2024). Among recent advancements, the generative recommendation paradigm, pioneered by models such as TIGER (Rajput et al., 2023), has emerged as a transformative frontier(Deldjoo et al., 2024; Li et al., 2023b; 2024). This approach redefines the retrieval process by training generative models to autoregressively decode target item identifiers, leveraging expressive models to capture the intricate and evolving nature of user preferences.

However, the practical effectiveness of generative recommendation is often undermined by a fundamental yet overlooked vulnerability: its sensitivity to the pervasive high-frequency sequential noise inherent in raw user interaction data. In real-world systems, user-item interactions are inherently noisy, a phenomenon that can bring challenges in two distinct ways: *semantic noise* and *high-frequency sequential noise*, as demonstrated in Figure 1.

- *Semantic noise* can arise from incomplete or misleading item data, such as missing brand information or misleading category details shown in the left of Figure 1, which can degrade the quality of semantically meaningful item representations or identifiers (Li et al., 2017). This corrupted foundational data can lead to inaccurate analysis and decision-making.

- *High-frequency sequential noise* exists within user behavior sequences as "soft noise" or "accidental clicks" that do not reflect genuine, long-term user interests (Du et al., 2023; Wang et al., 2021). As

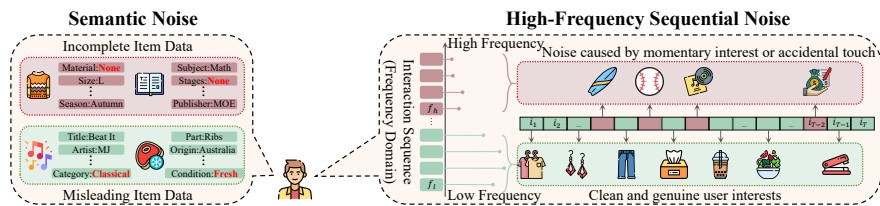

Figure 1: Illustration of our motivations. Left Panel: Semantic noise caused by incomplete or erroneous meta data, leading to difficulty in clustering. Right Panel: High-frequency sequential noise resulting from inaccurate interest modeling induced by short-term interests or accidental interactions.

depicted on the right of Figure 1, a user who rarely listens to music might accidentally click on an album, which results in a behavior contrary to the long-term interest in groceries. These short-term, non-predictive behaviors are highly coupled with a user's true preference signals, yet they can mislead the attention mechanisms of generative models, leading to performance degradation.

Tackling both challenges, this paper introduces a paradigm shift that explicitly performs frequency-domain modeling on sequential signals to effectively isolate and suppress high-frequency sequential noise. We propose TONE (Two-stage Optimized deNoising for gEnerative recommendation), a principled generative framework built around a two-stage denoising strategy. In the first stage of item codebook construction, we apply ResGMM (Residual Gaussian Mixture Model) to better fit clustering boundaries, thereby alleviating *semantic noise* and establishing a robust foundation. In the second stage, we employ a learnable Gaussian kernel to filter high-frequency noise on the generative model side. Furthermore, we show that the self-attention mechanism is implicitly a low-pass filter. To explicitly model the frequency-domain, we redesign the attention mechanism with a novel residual frequency-domain attention that explicitly separates real and imaginary components. We then introduce a learnable matrix to counteract attention collapse while preserving expressiveness with theoretical evidence.

Empirical results demonstrate that TONE achieves new state-of-the-art performance over strong baselines on three widely used benchmarks, achieving particularly notable improvements on the Amazon Beauty dataset, with gains of 8.93% in Recall@20 and 8.33% in NDCG@20. We find that TONE's superior performance stems from its ability to effectively handle the implicit noise and data quality issues which remain critical challenges for existing generative frameworks.

The key contributions of this work are summarized as follows:

- **Novel Two-Stage Denoising Strategy**: We propose a principled generative recommendation framework, TONE, that introduces a new paradigm centered on explicit noise suppression. We present a novel *two-stage denoising strategy* that addresses both semantic noise in item representations and high-frequency sequential noise in user sequences.

- **New Frequency-Domain Components**: We introduce *ResGMM for codebook construction* and a *residual frequency-domain attention mechanism with learnable matrix*, which are designed to enhance model robustness and prevent attention collapse during the explicit modeling of frequency.

- **SOTA Empirical Performance**: We establish a new state-of-the-art across multiple benchmarks, demonstrating that explicit frequency-domain denoising is key to unlocking a new level of performance and robustness in generative recommendation.

## 2 RELATED WORK

**Generative Recommendation.** Generative recommendation has emerged as a promising paradigm in which recommender models directly generate item identifiers as outputs with the help of generative models (Rajput et al., 2023; Ren et al., 2024). Early work applied classical generative models, including variational autoencoders (VAE) (Shenbin et al., 2020; Cai & Cai, 2022), generative adversarial networks (GAN) (He et al., 2018; Guo et al., 2020; Wang et al., 2022b), and diffusion-based methods (Jiang et al., 2024; Wang et al., 2023), to learn the underlying distribution of user–item interactions and produce new recommendation samples. More recently, researchers have leveraged large pre-trained language models (PLMs) to formulate recommendation tasks as natural language generation. Methods such as P5 (Geng et al., 2022) and M6-Rec (Cui et al., 2022) recast next-item prediction as a sequence-to-sequence task via prompts, aligning the recommendation objective with PLM pre-training. Follow-up work employs parameter-efficient fine-tuning or instruction tuning to inject linguistic knowledge into recommendation models (Lin et al., 2025; Bao et al., 2023).

A crucial challenge is how to represent each item as a unique token sequence that PLMs can understand. Early solutions used names or random numeric IDs, often with poor transferability (Geng et al., 2022; Cui et al., 2022). To address this, P5-ID explores identifier assignment strategies (Hua et al., 2023), ColaRec learns semantic tokens by capturing the collaborative signals between items (Wang et al., 2024), and TIGER encodes item embeddings into discrete codewords via vector quantization (Rajput et al., 2023). These works highlight the importance of bridging PLMs and recommendation through semantic identifiers, but none of them specifically considers the semantic noise naturally existing in the item metadata. On the contrary, our TONE deliberately uses ResGMM to construct healthier clustering boundaries through alleviating the semantic noise.

**Frequency-based Sequential Recommendation.** Beyond time-domain modeling (Fang et al., 2020; Boka et al., 2024), frequency-domain methods enhance sequential models by identifying periodic or noisy patterns. FMLP-Rec integrates a learnable frequency filter to de-noise user behavior (Zhou et al., 2022), while FEARec employs a frequency ramp structure to jointly learn short- and long-term information (Du et al., 2023). FamouSRec (Zhang et al., 2025a) proposes a mixture of heterogeneous experts to capture diverse behavioral patterns across different frequency ranges. Such frequency-based enhancements improve prediction accuracy, but most adopt a uniform strategy across users, ignoring the short-term, non-predictive behaviors as noises in user history. In contract, our proposed TONE explicitly models sequences from the perspective of frequency domain by redesigning the residual frequency-domain attention mechanism.

## 3 PRELIMINARIES

**Problem Formulation.** We consider a standard sequential recommendation setting, where each user $u$ is associated with an interaction history $S_u = [i_1, i_2, \ldots, i_T]$, with $i_t$ denoting the item interacted at time step $t$. The goal of generative recommendation is to predict the next item $i_{T+1}$ by modeling the conditional distribution:

$$P(i_{T+1} \mid S_u) = \prod_{t=1}^{T} P(i_t \mid i_{<t}), \tag{1}$$

where auto-regressive decoding treats item identifiers as tokens in a vocabulary.

**Frequency-Domain Perspective.** A user interaction sequence $x(t), t \in [1, T]$ in the time domain can be projected into the frequency domain using the Discrete Fourier transform (DFT):

$$X(f) = \sum_{t=1}^{T} x(t) e^{-j2\pi(f-1)(t-1)/T}, \quad f \in [1, T]. \tag{2}$$

where $X(f)$ denotes the complex-valued spectrum at frequency index $f$, and $j$ is the imaginary unit. The original sequence is then reconstructed by the inverse DFT (IDFT):

$$x(t) = \frac{1}{T} \sum_{f=1}^{T} X(f) e^{j2\pi(f-1)(t-1)/T}. \tag{3}$$

where the factor $\frac{1}{T}$ ensures perfect reconstruction of the time-domain sequence from its frequency spectrum.

## 4 METHODOLOGY

### 4.1 OVERALL FRAMEWORK

We propose TONE, a novel generative paradigm that addresses semantic noise in codebook construction and high-frequency sequential noise in sequence modeling within a unified two-stage framework (Figure 2). The model auto-regressively generates target semantic identifiers, which are mapped back to items via codebook lookup, producing the final recommendation results.

### 4.2 STAGE I: CODEBOOK CONSTRUCTION WITH RESGMM

In the codebook construction stage, learnable codebooks are used to convert items from dense embedding vectors to discrete semantic labels, which is proposed by TIGER (Rajput et al., 2023). As a widely used and classic approach for codebook construction, Residual Quantized Variational

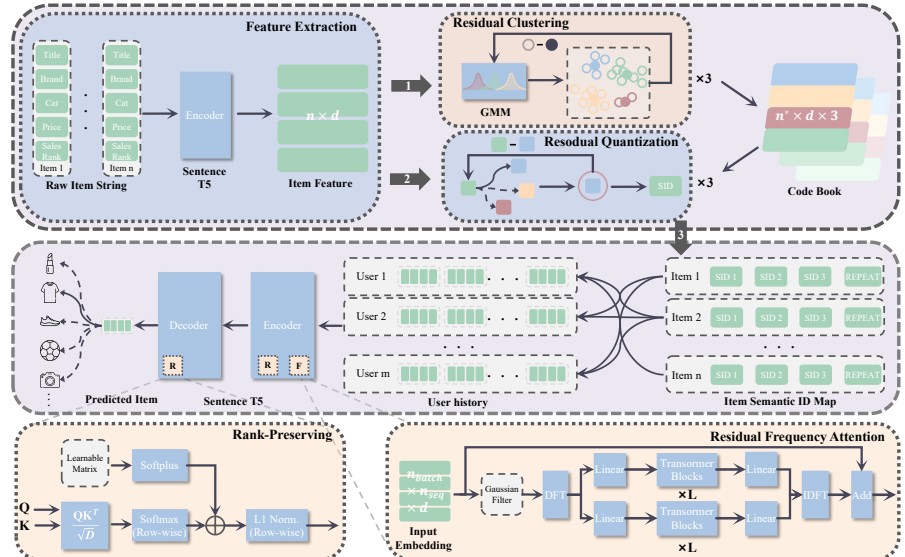

Figure 2: The overall framework of TONE. Our approach introduces a two-stage enhancement: (i) in the codebook generation stage, a ResGMM module is employed to suppress semantic noise; (ii) in the sequence modeling stage, a residual frequency-domain attention with separated real and imaginary components, together with a rank-preserving module, is applied to filter high-frequency sequential noise while strengthening the modeling of user interests.

Autoencoders (RQ-VAE) nonetheless suffer from a fundamental limitation in the marked under-utilization of shallow codebooks. Our empirical evidence in the Beauty dataset demonstrates that the first-level codebook typically engages fewer than 30% of its entries. Such sparsity leads to coarse and redundant representations, which restricts the model's discrimination over semantically heterogeneous items. In practice, incomplete or unreliable item metadata further exacerbates this challenge. Therefore, we propose a *Residual Gaussian Mixture Model* (ResGMM) scheme to enhance codebook utilization while reducing the effects of semantic noise.

ResGMM can be decoupled into two stages: clustering and residual quantization. In the clustering stage, GMM is applied to model data with multiple Gaussians, estimating parameters via likelihood maximization to softly assign data to high-posterior clusters. Given a text corpus $\mathcal{S}_{\text{raw}} = [\text{item}_1, \text{item}_2, \ldots, \text{item}_n]$ containing $n$ items, we employ a pre-trained `sentence-t5` encoder to map them into dense embeddings, which results in semantically informed vectors $\mathcal{S}_{\text{emb}} = [\boldsymbol{s}_1, \boldsymbol{s}_2, \ldots, \boldsymbol{s}_n] \in \mathbb{R}^{n \times d}$, where $d$ is the embedding dimension. Then for each item embedding $\boldsymbol{s}$, the label's probability is calculated as follows:

$$p(\boldsymbol{s}|\boldsymbol{\theta}) = \sum_{p=1}^{n^*} \pi_p \mathcal{N}(\boldsymbol{s}|\boldsymbol{\mu}_p, \boldsymbol{\Sigma}_p), \tag{4}$$

where $n^*$ denotes the number of Gaussian components, $\boldsymbol{\theta} = \{\pi_1, \ldots, \pi_{n^*}; \boldsymbol{\mu}_1, \ldots, \boldsymbol{\mu}_{n^*}; \boldsymbol{\Sigma}_1, \ldots, \boldsymbol{\Sigma}_{n^*}\}$ represents the set of model parameters; $\pi_p$ is the mixing coefficient of the $p$-th component, satisfying $\sum_{p=1}^{n^*} \pi_p = 1$ and $0 \leq \pi_p \leq 1$; $\mathcal{N}(\boldsymbol{s}|\boldsymbol{\mu}_p, \boldsymbol{\Sigma}_p)$ stands for the $p$-th multivariate Gaussian distribution, which is defined as:

$$\mathcal{N}(\boldsymbol{s}|\boldsymbol{\mu}_p, \boldsymbol{\Sigma}_p) = \frac{1}{(2\pi)^{d/2}|\boldsymbol{\Sigma}_p|^{1/2}} \exp\left\{-\frac{1}{2}(\boldsymbol{s} - \boldsymbol{\mu}_p)^\top \boldsymbol{\Sigma}_p^{-1}(\boldsymbol{s} - \boldsymbol{\mu}_p)\right\}, \tag{5}$$

where $\boldsymbol{\mu}_p \in \mathbb{R}^d$ denotes the mean vector, and $\boldsymbol{\Sigma}_p \in \mathbb{R}^{d \times d}$ is the covariance matrix.

For residual quantization, the input is initialized as $\boldsymbol{res}_1 = \mathcal{S}_{\text{emb}}$. Passing $\boldsymbol{res}_1$ through GMM produces the first codebook with $n^*$ cluster centroids: $\boldsymbol{C}_1 = [\boldsymbol{q}_1^1, \boldsymbol{q}_2^1, \ldots, \boldsymbol{q}_{n^*}^1] \in \mathbb{R}^{n^* \times d}$. For each $\boldsymbol{s}_i$, we compute the cosine similarity with all codewords in $\boldsymbol{C}_1$ to identify its nearest neighbor $\boldsymbol{q}_j^1$ and its corresponding index $\text{ind}_i^1$, and form the quantization set $\boldsymbol{Q}_1$. The residual is updated and used as input to the next stage through $\boldsymbol{res}_2 = \boldsymbol{res}_1 - \boldsymbol{Q}_1$. This recursive procedure is repeated three times, producing three residual codebooks $\boldsymbol{C}_1, \boldsymbol{C}_2, \boldsymbol{C}_3$. As a result, each item is represented by a

sparse 3-tuple of indices $(\text{ind}_i^1, \text{ind}_i^2, \text{ind}_i^3)$. We further adopt the duplication-bit strategy proposed by (Rajput et al., 2023) to prevent codebook collisions. Finally, we obtain the item semantic identifier set $\mathcal{S}_{\text{sid}} = [\boldsymbol{sid}_1, \boldsymbol{sid}_2, \ldots, \boldsymbol{sid}_n] \in \mathbb{R}^{n \times 4}$, where $\boldsymbol{sid}_i$ denotes the semantic id of item $i$.

By leveraging GMM for soft clustering with adaptive covariances, ResGMM achieves balanced codebook utilization (from <30% to >95%, as observed in our experiment) while its residual hierarchy captures coarse-to-fine semantics across levels. Moreover, the combination of EM-based regularization and residual decorrelation reduces overfitting to noisy metadata, producing more robust and discriminative semantic identifiers that reinforce generative modeling for recommendation.

### 4.3 STAGE II: GENERATIVE SEQUENTIAL MODELING WITH FREQUENCY ENHANCEMENT

In the sequential modeling stage, user interaction histories are mapped into sequences of semantic identifiers and decoded autoregressively by a frequency-enhanced Transformer-based generative model. User behavior sequences often include high-frequency noise (*e.g.*, spurious clicks) that impairs stable interest modeling. Typically, long-term user interest resides in low-frequency components, while short-term, non-predictive behavior is concentrated in high frequencies (Du et al., 2023; Wang et al., 2021). Existing works have demonstrated that the self-attention acts as a low-pass filter, since the self-attention calculates the weighted average of the value vectors of tokens, as proved in the following Lemma 1:

**Lemma 1.** *(Wang et al., 2022a) We formulate the Self-Attention (SA) module as below:*

$$\text{SA}(\boldsymbol{X}) = \text{softmax}\left(\frac{\boldsymbol{X}\boldsymbol{W}_Q(\boldsymbol{X}\boldsymbol{W}_k)^T}{\sqrt{d}}\right)\boldsymbol{X}\boldsymbol{W}_V, \tag{6}$$

*where $\boldsymbol{W}_k \in \mathbb{R}^{d \times d_k}, \boldsymbol{W}_Q \in \mathbb{R}^{d \times d_q}, \boldsymbol{W}_V \in \mathbb{R}^{d \times d}$ are the key, query, and value weight matrices, $\sqrt{d}$ denotes a scaling factor, and $\text{softmax}(\cdot)$ operates on $\boldsymbol{X}$ row-wisely. Let $\boldsymbol{A} = \text{softmax}(\boldsymbol{P})$, where $\boldsymbol{P} \in \mathbb{R}^{n \times n}$. Then $\boldsymbol{A}$ must be a low-pass filter. For all $\boldsymbol{z} \in \mathbb{R}^n$, $\lim_{t \to \infty} \|\mathcal{HC}[\boldsymbol{A}^t \boldsymbol{z}]\|_2 / \|\mathcal{DC}[\boldsymbol{A}^t \boldsymbol{z}]\|_2 = 0$, where $\mathcal{DC}[\boldsymbol{z}] = \tilde{\boldsymbol{z}}_{dc} \boldsymbol{f}_1 \in \mathbb{C}^n$ is the Direct-Current (DC) component of signal $\boldsymbol{z}$, and $\mathcal{HC}[\boldsymbol{z}] = [\boldsymbol{f}_2, \cdots, \boldsymbol{f}_n] \tilde{\boldsymbol{z}}_{hc} \in \mathbb{C}^n$ the high-frequency component.*

The proof of Lemma 1 can be referred to (Wang et al., 2022a). Lemma 1 reveals that self-attention behaves as a low-pass filter that only preserve the DC component, while diminishing the remaining high-frequency component. Although the desired low-pass pattern excels at modeling long-term, low-frequency patterns, this implicit modeling nature of self-attention module might not be capable of resisting the high-frequency sequential noise in a user's behavioral patterns. Therefore, this architectural rigidity constitutes a form of systemic flaw that leads to suboptimal performance.

To alleviate the implicit modeling of self-attention and exploit advantage in the frequency-domain explicitly, we introduce a frequency-aware architecture: (1) *Adapted Gaussian Filtering* (AGF) dynamically suppresses noise via learnable band-pass filtering; (2) *Complex Residual Frequency Attention* (CRFA) performs phase-sensitive modulation in the spectral domain; (3) *Rank-Preserving Matrix Learning* (RPML) maintains the embedding similarity structure post-filtering. Together, they enable principled spectral shaping, enhancing robustness without sacrificing representational fidelity.

#### 4.3.1 RESIDUAL FREQUENCY-DOMAIN ATTENTION MECHANISM

Our model builds upon an encoder-decoder architecture like `sentence-t5`. Given $X \in \mathbb{R}^{n_{\text{batch}} \times n_{\text{seq}} \times d}$ as input, where $n_{\text{batch}}$ is the number of user sequences per batch, $n_{\text{seq}}$ is the user sequence length, we design three core components as below:

**Adaptive Gaussian Filtering (AGF).** We apply a one-dimensional convolution with a learnable Gaussian kernel to suppress high-frequency noise. The kernel is defined as:

$$g(m; \sigma) = \frac{1}{\sqrt{2\pi}\sigma} \exp\left(-\frac{m^2}{2\sigma^2}\right), \quad m \in \{-k/2, \ldots, k/2\}, \tag{7}$$

where $k$ denotes the kernel size, $\sigma$ is a learnable parameter controlling the standard deviation, and $m$ represents the position index within the kernel. The convolution operation is expressed as:

$$X_{\text{filtered}}[:, i, :] = \sum_{m=-k/2}^{k/2} X[:, i-m, :] \cdot g(m; \sigma), \tag{8}$$

where $i$ denotes the sequence position index, yielding filtered embeddings $X_{\text{filtered}} \in \mathbb{R}^{n_{\text{batch}} \times n_{\text{seq}} \times d}$. The learnable $\sigma$ enables adaptive high-frequency suppression, resulting in cleaner user interest.

**Complex Residual Frequency Attention (CRFA).** Following AGF, we apply the Complex Residual Frequency Attention (CRFA) to convert the filtered time-domain signal into its frequency spectrum. The filtered time-domain signal $X_{\text{filtered}}$ is transformed into the frequency domain using DFT to obtain $X_{\text{freq}} \in \mathbb{R}^{n_{\text{batch}} \times \lceil (n_{\text{seq}}+1)/2 \rceil \times d}$. After merging the last two dimensions via the Flatten$(\cdot)$ operation and achieving dimension alignment through the Linear$(\cdot)$ layer, the real and imaginary parts of $X_{\text{freq}}$ are separated. These separated components are then fed into independent $L$-layer enhanced Transformer Blocks (TrmBlock) for decoupled attention learning on the phase and amplitude of the frequency-domain signal, resulting in the outputs $\text{Re}^l$ (learned real part) and $\text{Im}^l$ (learned imaginary part).

To maximize the utilization of mutual information, the IDFT is applied to convert $\text{Re}^l$ and $\text{Im}^l$ back to the time domain. Finally, a residual connection is established between this reconstructed time-domain signal and the original $X_{\text{filtered}}$, yielding the user's historical context with high-frequency noise filtered out and time-frequency fusion achieved. The detailed description of CRFA module is illustrated in Algorithm 1.

---

**Algorithm 1** Complex Residual Frequency Attention (CRFA) module.

---

**Input:** $X_{\text{filtered}} \in \mathbb{R}^{n_{\text{batch}} \times n_{\text{seq}} \times d}$, the number of stacked Transformer blocks $L$
**Output:** $X_{\text{out}} \in \mathbb{R}^{n_{\text{batch}} \times n_{\text{seq}} \times d}$
1: $X_{\text{freq}} \leftarrow \text{DFT}(X_{\text{filtered}})$
2: $\text{Re}^l \leftarrow \text{Linear}(\text{Flatten}(\text{Re}(X_{\text{freq}})))$
3: $\text{Im}^l \leftarrow \text{Linear}(\text{Flatten}(\text{Im}(X_{\text{freq}})))$
4: **for** $l = 1$ **to** $L$ **do**
5: $\quad \text{Re}^l \leftarrow \text{TrmBlock}(\text{Re}^{l-1})$
6: $\quad \text{Im}^l \leftarrow \text{TrmBlock}(\text{Im}^{l-1})$
7: **end for**
8: $\tilde{\text{Re}} \leftarrow \text{Linear}(\text{Re}^L)$
9: $\tilde{\text{Im}} \leftarrow \text{Linear}(\text{Im}^L)$
10: $\hat{X}_{\text{freq}} \leftarrow \text{IDFT}(\tilde{\text{Re}} + j \cdot \tilde{\text{Im}})$
11: $X_{\text{out}} \leftarrow X_{\text{filtered}} + \hat{X}_{\text{freq}}$
12: **return** $X_{\text{out}}$

---

In this way, our method explicitly models frequency domain, which allows for decoupled learning of phase and magnitude information. By integrating time-frequency domain information and providing richer attention expressions, we can better model the representation filtered by AGF.

### 4.3.2 RANK-PRESERVING MATRIX LEARNING

The sparsity inherent in frequency-domain representations often drives attention matrix toward low-rank, thereby limiting their capacity to capture complex dependency patterns. This phenomenon becomes particularly pronounced in frequency-domain modeling: when high-frequency noise is effectively suppressed, the consequent rank reduction in attention matrices creates an information bottleneck that severely compromises the model's ability to model long-term user interests.

To address this fundamental limitation, we introduce a novel rank-enhancement mechanism. Within each Transformer block, we incorporate a learnable full-rank matrix $M \in \mathbb{R}^{n_{\text{seq}} \times n_{\text{seq}}}$, constructed via diagonal dominance with random perturbations to ensure full-rank properties. This matrix is directly additive to the original attention scores:

$$A' = A + \alpha * \text{Softplus}(M), \tag{9}$$

where $A$ denotes the attention score matrix generated by the CRFA module, $M$ is initialized with small-variance Gaussian noise ($\mathcal{N}(0, 0.01)$) and optimized as a learnable parameter, $\text{Softplus}(\cdot)$ ensures positive entries, and $\alpha$ is a learnable matrix weight controls the intensity of full rank matrix.

To further demonstrate that the introduced rank-preserving matrix could effectively expand the rank of original attention matrix, we provide the theoretical analysis as below:

**Lemma 2.** *(Yue et al., 2025) Let $A$ and $B$ be two matrices of the same size $N \times N$. The rank of their sum satisfies the following bounds:*

$$|\text{rank}(A) - \text{rank}(B)| \leq \text{rank}(A + B) \leq \text{rank}(A) + \text{rank}(B) \tag{10}$$

The proof can be found in (Yue et al., 2025). The original attention matrix $A$ often exhibits a low rank, while the learned matrix $M$ is nearly full-rank. Therefore, Lemma 2 reveals that the combined matrix $A' = A + M$ generally achieves a higher rank.

This mechanism enables the model to effectively learn intrinsic frequency-domain correlations, thereby mitigating overfitting issues caused by frequency-domain sparsity while reinforcing the benefits of high-frequency denoising.

### 4.4 TRAINING AND RECOMMENDATION

**Training.** The training objective follows the generative recommendation paradigm, where the model learns to autoregressively predict the next item identifier conditioned on the historical sequence. Formally, given a user interaction sequence $S_u = \{\boldsymbol{sid}_1, \boldsymbol{sid}_2, \ldots, \boldsymbol{sid}_T\}$, each item $sid_t$ is represented by a tuple of semantic identifiers $sid_t = (\text{ind}_t^1, \text{ind}_t^2, \text{ind}_t^3, \text{ind}_t^4)$, derived from the codebook construction, the model is optimized to maximize the log-likelihood of sequential generation:

$$\mathcal{L} = \sum_{u=1}^{|\mathcal{U}|} \sum_{t=1}^{T} \log P_\theta\big(\text{ind}_t^1, \text{ind}_t^2, \text{ind}_t^3, \text{ind}_t^4 \mid \text{ind}_{<t}^{1:4}, \mathbf{Attr}\big), \tag{11}$$

where $\mathbf{Attr}$ denotes the associated item attributes and $\theta$ is the model parameters.

**Recommendation.** At inference time, the trained model autoregressively decodes the semantic codes of the next item using beam search with size $B = 30$. The final predicted item identifier is reconstructed from the decoded semantic codes, and the ranked recommendation list is obtained as

$$\hat{y}_t = \arg \max_{\boldsymbol{sid} \in \mathcal{I}} P_\theta(\boldsymbol{sid} \mid \text{ind}_{<t}^{1:4}, \mathbf{Attr}), \tag{12}$$

where $\mathcal{I}$ denotes the candidate item set. The top-$K$ items from this ranked list are returned for evaluation under HR@K and NDCG@K.

## 5 EXPERIMENTS

### 5.1 EXPERIMENTAL SETTINGS

**Datasets.**

To evaluate the effectiveness of our method, we conduct experiments on three benchmark datasets: Beauty[1] from Amazon Review Data 2014, Software[2] from Amazon Review Data 2018 and LastFM[3] from HetRec 2011 repository. The Amazon datasets capture user–item interactions enriched with metadata (title, ID, category and brand), while LastFM reflects user listening behaviors with associated artist names and tags. The detailed description and statistics can be found in the Appendix.

**Baseline Models.** Here, we compare our approach with four collaborative filtering (CF) methods, six recently developed LLM-based RecSys, and four widely-used generative recommendation couterparts. The detailed description of the baseline models can be found in the Appendix.

**Evaluation Settings.** We evaluate model performance using two standard metrics: Hit Ratio (HR@K) and Normalized Discounted Cumulative Gain (NDCG@K) (Järvelin & Kekäläinen, 2017), both of which assign higher scores to better recommendations. Results are reported as averages across all test users. The cutoff values of K are set to 10 and 20, with K = 10 adopted as the default in ablation studies and parameter analyses. The implementation details can be referred to the Appendix.

### 5.2 OVERALL PERFORMANCE

Table 1 reports the performance of collaborative filtering, LLM-based, and sequential recommendation methods across three benchmarks. Traditional CF (*e.g.*, MF, LightGCN) and sequential methods (*e.g.*, GRU4Rec, SASRec) generally underperform, reflecting their limited capacity to capture side information and long-term dependencies. LLM-based recommenders (*e.g.*, CoLLM, LlaRA, TokenRec, DeftRec) achieve moderate gains over CF baselines, while advanced sequential models such as TIGER exhibit stronger improvements, particularly on sparse datasets like LastFM. Our method consistently outperforms all baselines, achieving new state-of-the-art results on every dataset and evaluation metric. Compared with TIGER, the strongest competitor, our model attains notable gains, including +23.83% on NDCG@10 for Software and over +14.23% on NDCG@10 for LastFM.

The superior performance of our approach stems from two key design choices. First, the two-stage denoising strategy effectively suppresses semantically ambiguous noise and high-frequency sequential

---

[1]https://cseweb.ucsd.edu/ jmcauley/datasets/amazon/links.html
[2]https://nijianmo.github.io/amazon/index.html
[3]https://grouplens.org/datasets/hetrec-2011/

Table 1: Performance comparison across three representative approaches (NDCG@K denoted as NG@K). The best performance is marked as **bold**, while the second best results are underlined.

| Model | Software | | | | Beauty | | | | LastFM | | | |
|---|---|---|---|---|---|---|---|---|---|---|---|---|
| | HR@10 | HR@20 | NG@10 | NG@20 | HR@10 | HR@20 | NG@10 | NG@20 | HR@10 | HR@20 | NG@10 | NG@20 |
| MF | 0.1099 | 0.1570 | 0.0580 | 0.0702 | 0.0369 | 0.0585 | 0.0192 | 0.0217 | 0.0297 | 0.0389 | 0.0175 | 0.0218 |
| LightGCN | 0.1177 | 0.1702 | 0.0633 | 0.0763 | 0.0400 | 0.0616 | 0.0219 | 0.0253 | 0.0312 | 0.0458 | 0.0196 | 0.0233 |
| GTN | 0.1184 | 0.1686 | 0.0608 | 0.0758 | 0.0408 | 0.0642 | 0.0228 | 0.0256 | 0.0358 | 0.0535 | 0.0208 | 0.0266 |
| LTGNN | 0.1229 | 0.1686 | 0.0648 | 0.0787 | 0.0416 | 0.0627 | 0.0223 | 0.0259 | 0.0378 | 0.0532 | 0.0220 | 0.0262 |
| P5 | 0.1358 | 0.1683 | 0.0723 | 0.0819 | 0.0411 | 0.0594 | 0.0236 | 0.0258 | 0.0386 | 0.0555 | 0.0190 | 0.0225 |
| POD | 0.1345 | 0.1688 | 0.0715 | 0.0798 | 0.0414 | 0.0606 | 0.0227 | 0.0254 | 0.0402 | 0.0672 | 0.0219 | 0.0268 |
| CoLLM | 0.1362 | 0.1707 | 0.0723 | 0.0824 | 0.0429 | 0.0609 | 0.0237 | 0.0266 | 0.0468 | 0.0732 | 0.0228 | 0.0305 |
| LlaRA | 0.1361 | 0.1640 | 0.0717 | 0.0810 | 0.0438 | 0.0619 | 0.0226 | 0.0273 | 0.0489 | 0.0755 | 0.0238 | 0.0311 |
| TokenRec | 0.1469 | 0.1735 | 0.0797 | 0.0858 | 0.0442 | 0.0631 | 0.0237 | 0.0283 | 0.0525 | 0.0827 | 0.0244 | 0.0325 |
| DeftRec | 0.1584 | 0.1791 | 0.0863 | 0.0917 | 0.0474 | 0.0682 | 0.0255 | 0.0296 | 0.0539 | 0.0891 | 0.0252 | 0.0343 |
| GRU4Rec | 0.1051 | 0.1596 | 0.0593 | 0.0723 | 0.0379 | 0.0576 | 0.0197 | 0.0244 | 0.0329 | 0.0443 | 0.0183 | 0.0225 |
| SASRec | 0.1084 | 0.1652 | 0.0596 | 0.0756 | 0.0396 | 0.0591 | 0.0205 | 0.0247 | 0.0332 | 0.0455 | 0.0187 | 0.0230 |
| SSD4Rec | 0.1100 | 0.1653 | 0.0604 | 0.0781 | 0.0400 | 0.0605 | 0.0207 | 0.0248 | 0.0339 | 0.0465 | 0.0188 | 0.0231 |
| TIGER | 0.1570 | 0.2246 | 0.0877 | 0.1047 | 0.0558 | 0.0837 | 0.0304 | 0.0373 | 0.0851 | 0.1454 | 0.0408 | 0.0561 |
| TONE [Ours] | **0.1702** | **0.2261** | **0.1086** | **0.1227** | **0.0608** | **0.0911** | **0.0328** | **0.0404** | **0.0940** | **0.1493** | **0.0466** | **0.0606** |
| | +8.40% | +0.65% | +23.83% | +17.14% | +8.88% | +8.93% | +8.03% | +8.33% | +10.49% | +2.71% | +14.23% | +8.04% |

*Note:* Improvements in green are computed in comparison with the second best method TIGER.

noise, enabling cleaner and more faithful modeling of user interests. Second, the residual frequency-domain attention mechanism allows the model to exploit sparse spectral patterns, thereby capturing periodic user preferences more effectively. Together, these mechanisms contribute to the robustness and effectiveness of our approach across heterogeneous recommendation scenarios.

## 5.3 Ablation Study

**CodeBook Training.** To elucidate the influence of the underlying discretization paradigm on downstream recommendation effectiveness, we conduct a systematic evaluation of various clustering methodologies for codebook generation. This analysis builds on a three-level residual quantization framework, while keeping the codebook capacity aligned with the RQVAE baseline to ensure a fair comparison. We carefully choose the comparing methods and the detailed description of these methods are introduced in the Appendix. Our experimental results, presented in Table 2, indicate that most of the offline clustering-based strategies for codebook construction outperform the end-to-end trained RQVAE baseline, where the prefix "Res" indicates the use of residual quantization. It suggests that decoupling quantization from the primary training objective leads to more semantically coherent discrete representations. Among these methods, ResBi-Kmeans performs the worst, which may be due to its rigid balancing constraint. Such a restriction may disrupt intrinsic semantic structures by splitting cohesive conceptual groups or forcing unrelated items into the same cluster. In comparison, ResGMM achieves superior performance, surpassing all deterministic hard-assignment clustering methods. It highlights the strength of the probabilistic assignment mechanism, which preserves better semantic continuity of item representations and exhibits enhanced robustness to noise.

Table 2: Performance comparison of different clustering methods on the Beauty dataset.

| Method | Beauty | | | | | |
|---|---|---|---|---|---|---|
| | HR@5 | HR@10 | HR@20 | NDCG@5 | NDCG@10 | NDCG@20 |
| RQVAE | 0.0360 | 0.0558 | 0.0837 | 0.0239 | 0.0304 | 0.0373 |
| ResKmeans | 0.0380 | 0.0574 | 0.0866 | 0.0249 | 0.0311 | 0.0384 |
| ResKmeans++ | 0.0380 | 0.0575 | 0.0860 | 0.0243 | 0.0306 | 0.0378 |
| ResBi-Kmeans | 0.0334 | 0.0543 | 0.0800 | 0.0226 | 0.0293 | 0.0358 |
| ResSpectral | 0.0362 | 0.0584 | 0.0867 | 0.0235 | 0.0307 | 0.0378 |
| ResGMM | 0.0381 | 0.0584 | 0.0882 | 0.0250 | 0.0315 | 0.0390 |

Table 3: Ablation analysis of TONE under various methods on the Beauty dataset.

| Method | Beauty | | | | | |
|---|---|---|---|---|---|---|
| | HR@5 | HR@10 | HR@20 | NG@5 | NG@10 | NG@20 |
| w/o LM&AGF | 0.0293 | 0.0449 | 0.0694 | 0.0191 | 0.0241 | 0.0303 |
| w/o CRFA&AGF | 0.0354 | 0.0566 | 0.0859 | 0.0229 | 0.0297 | 0.0370 |
| w/o CRFA&LM | 0.0316 | 0.0480 | 0.0727 | 0.0210 | 0.0262 | 0.0325 |
| w/o AGF | 0.0343 | 0.0526 | 0.0769 | 0.0228 | 0.0286 | 0.0347 |
| w/o LM | 0.0342 | 0.0505 | 0.0755 | 0.0226 | 0.0279 | 0.0342 |
| w/o CRFA | 0.0323 | 0.0491 | 0.0751 | 0.0211 | 0.0265 | 0.0330 |
| TONE | **0.0397** | **0.0608** | **0.0911** | **0.0260** | **0.0328** | **0.0404** |

**Effectiveness of Proposed Modules.** We conducted a comprehensive ablation study on the Beauty dataset to evaluate the effectiveness of each proposed module. As shown in Table 3, the complete (TONE) achieves the highest performance, outperforming TIGER by 8.93% in HR@20 and 8.33% in NDCG@20, where CRFA=Complex Residual Frequency Attention, LM=learnable matrix, and AGF=Adapted Gaussian Filter. This result demonstrates that optimal performance arises when all modules are integrated, thereby validating the effectiveness of the overall design.

Furthermore, the results indicate that eliminating either one or two modules consistently causes a noticeable degradation in performance. Specifically, the learnable matrix (w/o CRFA&AGF) proved to be the most impactful standalone module, which enhances attention diversity and generalization. Similarly, configurations such as (w/o LM&AGF) or (w/o CRFA&LM) also suffered performance drops, which can be attributed to optimization instability caused by frequency-domain sparsity and semantic distortion from oversmoothing, respectively. The results further reveal that the components reinforce each other, as the integration of AGF and CRFA (w/o LM) clearly surpasses either com-

ponent alone. This finding substantiates our hypothesis that suppressing high-frequency sequential noise is essential for reliably modeling the periodic user interests captured by CRFA.

In summary, the results confirm that CRFA, LM, and AGF work as an integrated pipeline, rather than as simply additive components. Specifically, CRFA captures user interests with fine-grained resolution in the frequency domain, LM alleviates overfitting to dominant frequency patterns by enforcing full-rank representations, and AGF enhances reliability by filtering out high-frequency noise, allowing the model to focus on genuine user interest sequences.

## 5.4 SENSITIVITY TO HYPERPARAMETERS

To better understand the influence of critical hyperparameters in TONE, we conduct a sensitivity analysis of model performance w.r.t. the learnable matrix weight $\alpha$, the Gaussian filter's standard deviation $\sigma$, and the kernel size. As illustrated in Figure 3, HR@10 peaks at $\alpha = 0.4$ and $\sigma = 3.0$, a pattern that holds consistently across all three datasets, indicating reliable and generalizable hyperparameter choices. The results regarding the kernel size can be referred to the Appendix.

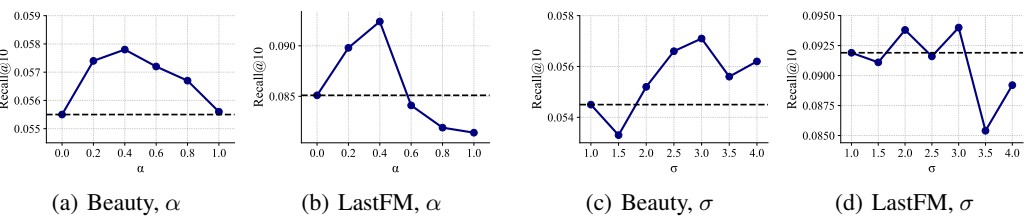

(a) Beauty, $\alpha$      (b) LastFM, $\alpha$      (c) Beauty, $\sigma$      (d) LastFM, $\sigma$

Figure 3: The effect of learnable matrix weight $\alpha$ and Gaussian Filter $\sigma$ under HR@10.

## 5.5 LEARNABLE MATRIX VALIDITY

We validated the full-rank enhancement of the attention matrix by visualizing attention patterns from a randomly sampled encoder head at the 30th epoch, contrasting the results with and without the learnable matrix. As shown in Figure 4, the integration of the learnable matrix increases the rank from 18 to 41 on Beauty and from 3 to 41 on LastFM, demonstrating its effectiveness in enriching attention expression diversity, enabling TONE to capture more complex user–item interaction patterns. The results highlight the advantage of spectral enhancement in sequential recommendation.

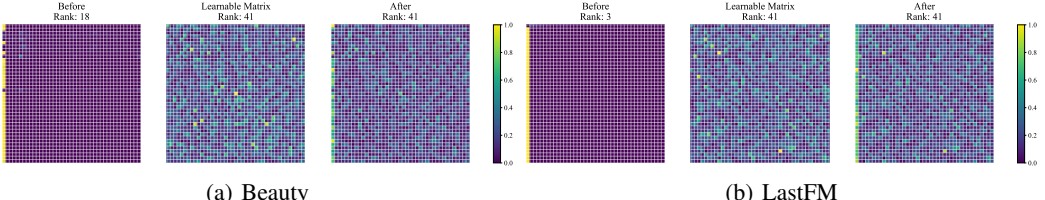

(a) Beauty            (b) LastFM

Figure 4: The visualization of the rank preservation effect of LM on different datasets.

## 6 CONCLUSION

This work confronts a fundamental challenge in generative recommendation: its vulnerability to high-frequency noise in user behavior sequences. We argue that robust recommendation requires explicit noise handling, not just implicit modeling. To this end, we introduce TONE, a framework that embeds frequency-domain denoising into its core. Our two-stage approach first stabilizes item representations via ResGMM, then filters sequential noise with a novel residual frequency-domain attention mechanism. The results are clear: TONE achieves new state-of-the-art performance across benchmarks. Its significant gains on Amazon Beauty (over 8% in key metrics) underscore that explicitly suppressing noise is a critical factor for next-generation recommenders.This work establishes frequency-domain denoising as a powerful principle for building more robust and accurate generative models. We believe this perspective opens new avenues for creating reliable recommendation systems.

ETHICS STATEMENT

This research employs exclusively publicly available datasets, which have been subjected to standard anonymization protocols. We have conducted a comprehensive analysis of potential biases and broader societal impacts inherent in the proposed model. To ensure full transparency and facilitate reproducibility, the complete source code and data necessary to replicate our findings will be made publicly available upon publication.

REPRODICIBILITY STATEMENT

We open-sourced all experimental code and data in an anonymous repository. All experimental results can be reproduced using the provided code.

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

Table 4: Summary statistics of the benchmark datasets.

| Dataset | User | Item | Interactions | Avg |
|---------|------|------|-------------|-----|
| Software | 1,351 | 778 | 9322 | 7.42 |
| Beauty | 22,363 | 12,101 | 278,641 | 7.88 |
| LastFM | 1091 | 3,685 | 52,670 | 16.62 |

## A  ADDITIONAL EXPERIMENTAL RESULTS

### A.1  DATASETS

Table 4 details the characteristics of these datasets. We apply uniform pre-processing by discarding users with fewer than five interactions, truncating sequences of more than 20 items to the most recent entries, and preserving chronological order. The concatenated item attributes are uniformly embedded using the pre-trained `sentence-t5`. For evaluation, we follow the leave-one-out protocol, holding out the last interaction for testing, the second-to-last for validation, and using the remainder for training.

### A.2  BASELINE MODELS

Here, we compare our approach with four collaborative filtering (CF) methods, six recently developed LLM-based RecSys, and four widely-used generative recommendation couterparts. The detailed description can be found below:

- **MF** (Koren et al., 2009): Introduces matrix factorization for collaborative filtering, incorporating implicit feedback to model user-item interactions.
- **LightGCN** (He et al., 2020): Simplifies GCN by removing nonlinearities and proposes a lightweight linear propagation scheme for efficient recommendation.
- **GTN** (Fan et al., 2022): Addresses spectral oversmoothing in GNNs via graph trend filtering and employs PAPR iteration for robust graph-based recommendation.
- **LTGNN** (Zhang et al., 2024): Proposes implicit graph modeling and efficient variance-reduced sampling to enhance the scalability of GNN-based recommender systems.
- **P5** (Geng et al., 2022): Frames recommendation as a text generation task using prompts and pre-trains a unified text-to-text transformer model.
- **POD** (Li et al., 2023a): Distills discrete prompts into continuous vectors and introduces task-alternated training for effective prompt tuning in recommendation.
- **CoLLM** (Zhang et al., 2025b): Mitigates collaborative signal deficiency in LLM-based recommenders by integrating traditional model embeddings and a two-stage tuning strategy.
- **LLaRA** (Liao et al., 2024): Addresses modality alignment challenges in LLM-based sequential recommendation via hybrid prompting and curriculum learning.
- **TokenRec** (Qu et al., 2025b): Proposes a masked vector-quantized tokenizer and generative retrieval framework for item recommendation with LLMs.
- **DeftRec** (Qu et al., 2025a): Introduces an additive continuous tokenizer with contrastive denoising diffusion for high-fidelity item representation in LLM-based recommendation.
- **GRU4Rec** (Hidasi & Karatzoglou, 2018): Pioneers the use of GRUs for session-based recommendation, introducing novel sampling strategies and ranking losses.
- **SASRec** (Kang & McAuley, 2018): Leverages self-attention with positional encodings and residual blocks for sequential recommendation.
- **SSD4Rec** (Qu et al., 2024): Designs a Mamba-based sequential model with masked sequence modeling and bidirectional state-space blocks for efficient long-sequence recommendation.
- **TIGER** (Rajput et al., 2023): Proposes generative retrieval with RQ-VAE tokenization and autoregressive Transformer decoding for end-to-end recommendation.

### A.3 IMPLEMENTATION DETAILS

For the codebook training stage, the number of initializations in the Gaussian Mixture is set to $n\_init = 3$, and a three-level codebook with sizes $[256, 256, 256]$ is adopted. For the model generation stage, we use a batch size of 256, with both encoder and decoder consisting of 4 layers. The multi-head attention module employs 6 heads, with input dimension 128, query–key–value dimension 64, and feed-forward layer dimension 1024. We set the beam size to 30 and dropout to 0.1. Optimization is performed using the Adam optimizer with a learning rate of $1 \times 10^{-4}$ and an early stopping strategy. For baselines and ablation studies, the low-pass Gaussian kernel is parameterized with $\sigma = 3.0$, kernel size $= 7$, and $\alpha = 0.4$. For hyperparameter ablations, we vary the Gaussian kernel parameter $\sigma \in \{1.0, 1.5, 2.0, 2.5, 3.0, 3.5, 4.0\}$, kernel size $\in \{7, 9, 11, 13, 15\}$, and the weighting coefficient $\alpha \in \{0.0, 0.2, 0.4, 0.6, 0.8, 1.0\}$ for the learning matrix summation.

### A.4 CODEBOOK TRAINING ABLATION STUDY

We carefully choose the codebook training methods for ablation study. The compared methods include: (i) K-means, which partitions the latent space via hard assignments based on Euclidean proximity; (ii) K-means++, a variant with a more robust initialization scheme to enhance cluster stability; (iii) Bi-Kmeans, which enforces a strict cardinality balance across clusters, potentially compromising semantic coherence; (iv) Spectral Clustering, which leverages the eigenspectrum of a graph affinity matrix to discern non-linear data structures; and (v) the Gaussian Mixture Model (GMM), which operates under a generative framework to perform probabilistic soft assignments.

### A.5 VISUALIZATION OF COARSE-GRAINED SEMANTIC IDENTIFIERS DISTRIBUTION

As illustrated in Figure 5, we visualize the distribution of original text categories on the Beauty Dataset over the first digit of semantic identifiers $[0, 1, 2, 3]$. In contrast, RQVAE exhibits a shifted pattern $[0, 1, 3, 4]$ due to underutilization of its codebook, particularly lacking entries corresponding to category $[2]$. Our results show that ResGMM achieves significantly more balanced and discriminative activations in the first-level codebook compared to the baseline, with broader coverage across semantic entries. This indicates that ResGMM effectively mitigates shallow codebook under-use by refining coarse-level semantic partitioning. The resulting fine-grained and well-distributed codes provide a more expressive representation, forming a strong foundation for downstream modeling and enabling precise capture of nuanced user interests even at early abstraction layers. We also observed ResGMM achieves more balanced codebook utilization (from <90% to >95%), which further demonstrate the effectiveness of ResGMM.

### A.6 MORE RESULTS ON THE SENSITIVITY TO HYPERPARAMETERS

As illustrated in Figure 6, unlike $\alpha$ and $\sigma$, the optimal kernel size depends on dataset characteristics, with 7 for Beauty and 15 for LastFM, consistent with the anticipated timescales of user interest evolution.

### A.7 COMPUTATIONAL EFFICIENCY

We compared the training and inference time per epoch of TIGER and TONE on three benchmark datasets, as shown in Figure 7. While TONE incurs some overhead from the learnable matrix, the added training delay is relatively minor (+23.8% on Beauty, +29.73% on LastFM) compared with the substantial performance gains it brings. Moreover, during inference, the learnable matrix is precomputed and remains fixed, resulting in only marginal overhead (+12.24% on Beauty, +13.57% on LastFM), which can be considered negligible in practical deployments. As a result, TONE maintains inference latency on par with TIGER across datasets, which ensures effective deployment in real-world scenarios.

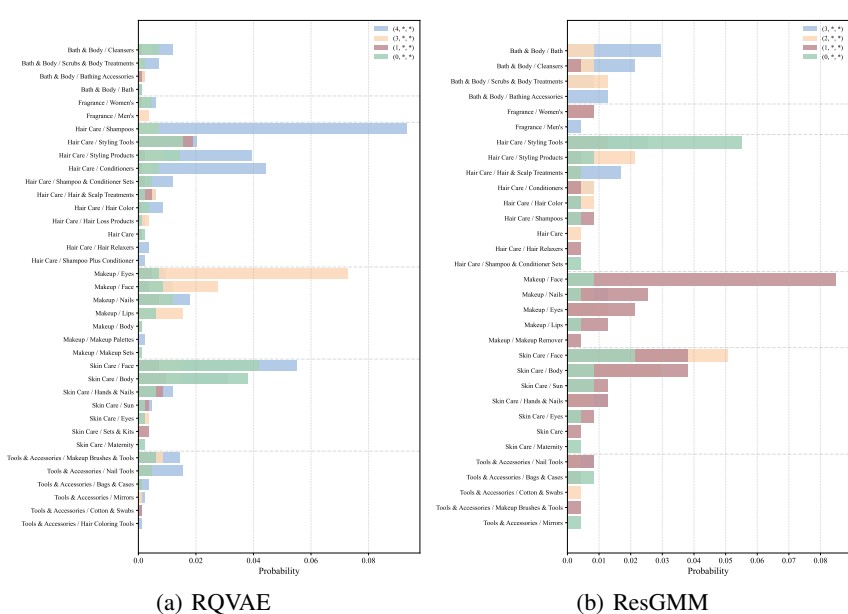

(a) RQVAE      (b) ResGMM

Figure 5: Visual comparison of semantic identifiers between RQVAE and ResGMM on ground truth category.

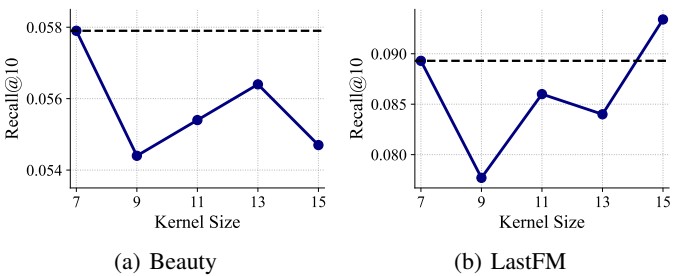

(a) Beauty      (b) LastFM

Figure 6: The effect of Gaussian Filter Kernel Size under HR@10 across three datasets.

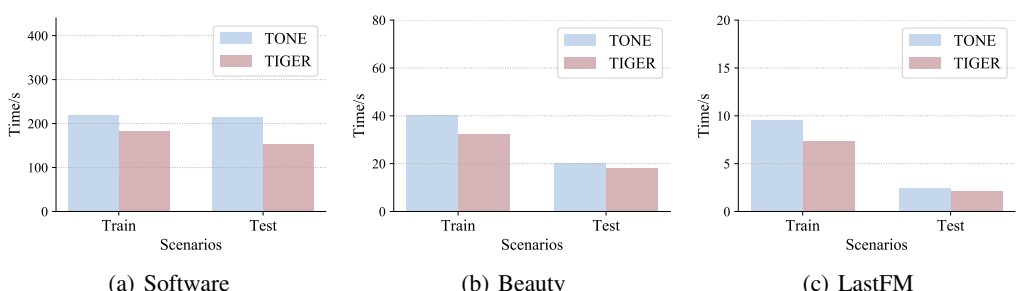

(a) Software      (b) Beauty      (c) LastFM

Figure 7: Comparison of the training and inference speeds of TONE and TIGER among three dataset

## B    USE OF LLMs

This article uses LLMs to reline certain aspects of writing logic and grammatical accuracy. In experiments, some portions of the code were generated with the assistance of LLMs. However, LLMs were not involved in the formulation of the core ideas or the overall structure of the manuscript.

