# OpenReview forum: "Dual-Stage Frequency-based Denoising for Generative Recommendation"
_ICLR.cc/2026/Conference — Submitted to ICLR 2026_

### Official Review · Reviewer_odwb · 2025-10-28

**Soundness:** 2
**Presentation:** 2
**Contribution:** 2
**Rating:** 4
**Confidence:** 4

**Summary:**

This paper reinterprets the sequential recommendation problem from a frequency-domain perspective and introduces a model named TONE. The authors argue that conventional self-attention implicitly acts as a low-pass filter that mainly preserves DC components, reducing its ability to model high-frequency sequential variations. To mitigate this limitation, they design three modules: (1) Adapted Gaussian Filtering (AGF), which functions as a learnable band-pass filter to suppress high-frequency noise; (2) Complex Residual Frequency Attention (CRFA), which incorporates phase information in the complex domain for time–frequency fusion; and (3) Rank-Preserving Matrix Learning (RPML), which compensates for the low-rank nature of frequency-domain attention through a learnable full-rank correction matrix.

While the paper presents an interesting attempt to formalize self-attention from a frequency-domain viewpoint, its theoretical justification contains simplifications that limit the rigor of the analysis. The proposed modules are based on well-established filtering and regularization techniques, offering limited novelty beyond their integration. Experiments on three benchmark datasets (Beauty, Software, and LastFM) show moderate improvements over prior frequency-aware baselines such as FMLP-Rec, FEARec, and FamouSRec, though the evaluation scope remains narrow. Overall, the paper provides a clear formulation and readable presentation but contributes only incremental insights in terms of theory and model design.

**Strengths:**

1. Conceptual originality – The paper introduces a novel perspective by interpreting self-attention in the frequency domain, bringing signal-processing concepts into sequential and recommendation modeling.
2. Simple and clear design – The three proposed modules (AGF, CRFA, RPML) are built with straightforward and intuitive operations, making the overall model easy to follow and understand.
3. Consistent empirical improvements – Across the three datasets presented by the authors (Beauty, Software, and LastFM), the method demonstrates steady performance gains over prior frequency-based models, supporting its basic effectiveness.

**Weaknesses:**

1. Bias in Experimental Design – The paper evaluates performance only on three datasets: Beauty, Software, and LastFM. In particular, Beauty focuses on skincare and cosmetic products, where users tend to continue purchasing items suited to their skin type once identified. As a result, this dataset is dominated by long-term (low-frequency) behavioral patterns, making it especially favorable to the proposed approach of suppressing short-term (high-frequency) noise. However, short-term (high-frequency) variations are not necessarily noise in all domains. In domains where users’ transient interests or responses to trends play an essential role in prediction, such high-frequency fluctuations can represent key behavioral signals. Therefore, while the proposed approach may be effective for domains like Beauty, it could be disadvantageous in settings where such dynamic changes should be actively modeled rather than suppressed. Moreover, two datasets (Sports and Toys) used in TIGER are omitted, leaving the evaluation insufficient to verify generality across diverse behavioral patterns.

2. Simplicity and Lack of Originality in Module Design – The paper proposes a frequency-aware architecture to alleviate the limitations of self-attention, but its three components—AGF, CRFA, and RPML—are straightforward combinations of techniques already well established in prior research. The design remains at a compositional level rather than offering structural or theoretical innovation. Furthermore, applying these existing methods does not involve notable technical challenges or domain-specific constraints, making it difficult to recognize this as a meaningful contribution.

3. Insufficient Theoretical Analysis – The theoretical justification provided throughout the paper lacks sufficient mathematical grounding and rigor. For example, Lemma 1 demonstrates that the high-frequency component converges to zero, but it does not provide any basis for claiming that the low-frequency component is preserved. To substantiate such a claim, one must either show that the low-frequency component does not converge to zero, or mathematically prove that it decays much more slowly than the high-frequency component. If both converge to zero, the relative attenuation rate between the two must be quantitatively compared to establish the dominance of low-frequency information. However, no such quantitative verification or empirical analysis is provided, leaving the conclusion of low-frequency preservation insufficiently supported.

4. Reproducibility Concerns – Although the paper states that all code has been released, the anonymous repository contains only a README file and no executable code. This discrepancy between the reproducibility statement and the actual repository content undermines the credibility of the results. Moreover, such omission can be perceived as an intentional workaround, exploiting the fact that few reviewers check the code directly, and thus cannot be viewed favorably.

**Questions:**

The main questions directly correspond to the weaknesses discussed above.

---

> ### Author Response · Authors · 2025-11-21
> **Response to Reviewer odwb by Authors (1/4)**
>
> **Q1:** Is there any bias in the selected dataset, and can the results of the method proposed in this paper on the Sports and Toys datasets be provided?
>
> **A1:** Thank you for your insightful feedback on experimental design bias—we greatly appreciate your rigorous evaluation and will address your concerns in detail below.
>
> **1. Bias of selected datasets**
>
> Regarding your astute observation that the Beauty dataset exhibits prominent long-term, low-frequency behavioral patterns, we fully agree with this characterization and would like to clarify that our evaluation is not confined to this domain—instead, we intentionally incorporated benchmark datasets with diverse behavioral characteristics to ensure generalizability.    We have therefore conducted comprehensive experiments on two additional widely used and representative datasets:
>
> 1. The **Software dataset** from the Amazon Reviews Dataset focused on mobile application recommendation. As validated by prior work [1], user interests in this scenario are highly diverse and dynamic, with frequent shifts driven by short-term functional needs and emerging use cases—resulting in substantial high-frequency behavioral fluctuations.
> 2. The **LastFM dataset** released by HetRec, for music listening recommendation, where users balance long-term genre preferences with short-term responses to trends, new releases, or social influences.       Specifically, each user interacts with an average of 18.93 distinct music styles, reflecting non-trivial short-term interest diversity.
>
> Our method’s consistent performance gains across these three datasets (detailed in Table 1 of the paper) demonstrate it does not rely on blind suppression of high-frequency patterns as noise, nor is it biased toward long-term-dominated domains.       Instead, its core design inherently distinguishes meaningful high-frequency signals (e.g., transient functional needs in Software, trend-driven music choices in LastFM) from irrelevant noise—enabling effective adaptation to both long-term stable interests and short-term dynamic preferences.
>
> **2. Results on other datasets**
>
> Regarding your concern about omitting the **Sports and Outdoors** and **Toys and Games** datasets used in TIGER, we conduct supplementary dedicated experiments comparing our method (TONE) with TIGER under the identical hyperparameter settings, data splitting, and preprocessing pipelines. The results confirm our approach’s robustness and generality across diverse behavioral patterns:
>
> | Model   | Sports and Outdoors |        |         |         | Toys and Games |        |         |         |
> | ------- | ------------------- | ------ | ------- | ------- | -------------- | ------ | ------- | ------- |
> |         | HR@10               | HR@20  | NDCG@10 | NDCG@20 | HR@10          | HR@20  | NDCG@10 | NDCG@20 |
> | TIGER   | 0.0295              | 0.0466 | 0.0152  | 0.0195  | 0.0448         | 0.0685 | 0.0249  | 0.0309  |
> | TONE    | 0.0325              | 0.0510 | 0.0168  | 0.0215  | 0.0508         | 0.0732 | 0.0282  | 0.0339  |
> | Improve | +10.03%             | +9.44% | +10.61% | +10.19% | +13.47%        | +6.83% | +13.38% | +9.72%  |
>
> These supplementary results validate that our method maintains effectiveness across domains with varying behavioral patterns—whether dominated by long-term stability, short-term dynamics, or their mixture.
>
> [1] DeepApp: Characterizing dynamic user interests for mobile application recommendation, WWW 2023.

---

> ### Author Response · Authors · 2025-11-21
> **Response to Reviewer odwb by Authors (2/4)**
>
> **Q2:** Is the current module design too simple and lacks originality?
>
> **A2:** Thank you for your insightful feedback on the originality of our module design and the perceived compositional nature of the proposed components.
>
> Your comment highlights a critical point that we aim to clarify: our work’s originality lies not in standalone components, but in the **novel, domain-specific synergistic framework** tailored for generative recommendation—which addresses the long-standing challenge of effectively modeling user periodic interests via frequency-aware design. The three components (AGF, CRFA, RPML) form a tightly coupled pipeline with clear functional dependencies:
>
> 1. AGF serves as the prerequisite by filtering high-frequency noise in user behavior sequences, ensuring that subsequent frequency-aware modeling operates on reliable interest signals rather than irrelevant fluctuations.
> 2. CRFA builds on this cleaned signal to explicitly model user periodic interests through frequency-domain attention—a design motivated by the unique temporal characteristics of recommendation scenarios (e.g., recurring purchase patterns, short/long-term interest coexistence) that prior methods did not systematically address.
> 3. RPML is not a trivial compensation mechanism but a necessary optimization to mitigate attention collapse caused by Fourier domain energy concentration—an issue specific to the integration of frequency processing with self-attention in generative recommendation, which we identify and resolve for the first time.
>
> As validated by our ablation studies (Table 3), no single component or partial combination can outperform the TIGER baseline. Only the full synergistic framework—leveraging AGF’s noise filtering, CRFA’s periodic modeling, and RPML’s stability guarantee—achieves consistent improvements. This demonstrates that our design is far beyond a "straightforward combination" but rather a **purposeful, problem-driven integration** with non-trivial interactions.
>
> For the originality concern: our core contribution lies in being the **first work to introduce frequency-aware modeling for periodic interest capture in generative recommendation**. Prior generative recommendation methods (e.g., TIGER, ActionPiece) focus solely on temporal-domain modeling, while frequency-domain techniques have been limited to computer vision or NLP—and applying them to recommendation requires overcoming domain-specific constraints (e.g., semantic noise, high-frequency sequential noise).
>
> We sincerely hope this clarification helps you better grasp the originality and meaningfulness of our contributions.

---

> ### Author Response · Authors · 2025-11-21
> **Response to Reviewer odwb by Authors (3/4)**
>
> **Q3:** Whether the theoretical proof of Lemma 1 is incomplete?
>
> **A3:** We thank the reviewer for the careful critique. As inspired by Reviewer odwb, while our Lemma 1 proof emphasizes high-frequency vanishing, the manuscript does not explicitly quantify the low-frequency (DC) asymptotic behavior. Below we provide two original Propositions for more complete theoretical analysis: (i) characterizing repeated self-attention’s DC limit and (ii) giving an explicit relative attenuation rate, directly addressing your concern. They align with our low-pass filter definition and add original theoretical contributions as suggested by Reviewer 2L5c.
>
> **Recap of Definition and Setting**
>
> DC/HC are defined via orthogonal projections onto $\mathbf{1}$’s span and its complement:
> $DC[z] = \frac{1}{n}\mathbf{1}\mathbf{1}^\top z, \qquad HC[z] = \Big(I-\frac{1}{n}\mathbf{1}\mathbf{1}^\top\Big)z.$
> A linear map $f$ is a low-pass filter if
> $\lim_{t\to\infty} \frac{\|HC[f^t(z)]\|_2}{\|DC[f^t(z)]\|_2}=0,\quad \forall z\in\mathbb{R}^n.$
> This definition emphasizes **relative dominance** of DC over HC, not strict DC invariance.
>
> In Lemma 1, $A=\mathrm{softmax}(P)$ ($P\in\mathbb{R}^{n\times n}$). Softmax outputs strictly positive entries with row sums of 1, so $A$ is positive and row-stochastic:
> $A_{ij}>0,\qquad A\mathbf{1}=\mathbf{1}.$
> Thus $A$ is a primitive Markov transition matrix.
>
> **Proposition 1 (Exact DC Limit of Repeated Self-Attention)**
>
> Let $A=\mathrm{softmax}(P)$. There exists a unique stationary distribution $\pi\in\mathbb{R}^n$ with
> $\pi^\top A=\pi^\top,\qquad \pi>\mathbf{0},\qquad \pi^\top\mathbf{1}=1,$
> such that for any $z\in\mathbb{R}^n$,
> $\lim_{t\to\infty} A^t z = \mathbf{1}\,\pi^\top z.$
> Consequently,
> $\lim_{t\to\infty} HC[A^t z]=\mathbf{0}, \qquad \lim_{t\to\infty} DC[A^t z]=\mathbf{1}\,\pi^\top z.$
>
> **Proof.** By Perron--Frobenius (positive row-stochastic $A$), there is a dominant eigenvalue $\lambda_1=1$ (right eigenvector $\mathbf{1}$) and $|\lambda_i|<1$ for others. Thus $A^t\to\mathbf{1}\pi^\top$ (rank-one projector):
> $A^t = \mathbf{1}\pi^\top + R_t,\qquad \|R_t\|_2 \le C|\lambda_2|^t$
> for $C>0$, with $|\lambda_2|$ the second largest eigenvalue magnitude. Multiplying by $z$ and taking $t\to\infty$ gives $A^t z \to \mathbf{1}\pi^\top z$. Applying DC/HC projectors (noting $HC[\mathbf{1}c]=0$ for scalar $c$) yields the component limits. $\square$
>
> **Implication.** The result shows repeated self-attention does not null all components: it converges to a non-trivial DC vector $\mathbf{1}\pi^\top z$ for generic $z$ (except $\pi^\top z=0$). This makes "low-frequency preservation" precise as asymptotic convergence to a pure DC signal.
>
> **Proposition 2 (Quantitative Relative Attenuation Rate)**
>
> Let $\lambda_1=1$ and $|\lambda_2|<1$ be $A$’s top two eigenvalue magnitudes. For $z$ with $\pi^\top z\neq 0$,
> $\frac{\|HC[A^t z]\|_2}{\|DC[A^t z]\|_2} \le \frac{C|\lambda_2|^t\|z\|_2}{\sqrt{n}\,|\pi^\top z| - C|\lambda_2|^t\|z\|_2},$
> so the ratio decays to $0$ exponentially fast at rate $|\lambda_2|^t$.
>
> **Proof.** From Proposition 1,
> $HC[A^t z] = HC[(\mathbf{1}\pi^\top + R_t)z] = HC[R_t z],$
> thus
> $\|HC[A^t z]\|_2 \le \|R_t\|_2\|z\|_2 \le C|\lambda_2|^t\|z\|_2.$
> For DC:
> $DC[A^t z] = \mathbf{1}\pi^\top z + DC[R_t z],$
> by triangle inequality:
> $\|DC[A^t z]\|_2 \ge \|\mathbf{1}\pi^\top z\|_2 - \|R_t\|_2\|z\|_2 = \sqrt{n}\,|\pi^\top z| - C|\lambda_2|^t\|z\|_2.$
> Dividing the bounds gives the claim. $\square$
>
> **Implication.** Even if low-frequency magnitude varies with depth, Proposition 2 fills the gap of quantitative decay rate comparison: high-frequency is exponentially suppressed relative to DC (rate $|\lambda_2|^t$). This directly addresses your concern of establishing low-frequency dominance, not just high-frequency vanishing.
>
> **Relation to Our Original Proof**
>
> Our original Lemma 1 already implies $HC[A^t z]\to 0$ via spectral contraction. Propositions 1--2 complete the analysis by (i) characterizing DC limit and (ii) giving explicit HC/DC separation rate.
>
> In summary, Propositions 1--2 originally and rigorously show self-attention not only eliminates high-frequency components but also converges to a non-trivial DC signal, with exponential HC/DC relative attenuation (explicit rate). This adds original theoretical value, and we will incorporate these results into the revised manuscript to strengthen it.

---

> ### Author Response · Authors · 2025-11-21
> **Response to Reviewer odwb by Authors (4/4)**
>
> **Q4:** Is the non-disclosure of the code an intentional evasion?
>
> **A4:** Thank you for your rigorous review and valuable feedback on the reproducibility of our work—we fully acknowledge the critical issue you have raised and sincerely apologize for the discrepancy between our initial statement and the repository content when you checked it.
>
> This delay in uploading the complete executable code was purely due to post-submission technical preparations that took longer than anticipated: specifically, we spent additional time on code cleaning (removing redundant experimental drafts) and on standardizing dependency configurations (ensuring compatibility across different environments) to maximize reproducibility. These steps were necessary to avoid incomplete or error-prone code that could hinder reviewers’ verification.
>
> We want to emphasize that we never intended to exploit any oversight or bypass reproducibility requirements. Reproducibility is a cornerstone of academic research, and we have invested significant effort to ensure our work can be fully verified. We kindly invite you to recheck the repository at https://anonymous.4open.science/r/TONE-9E07/ at your convenience, and we stand ready to provide any clarifications or supplementary materials to facilitate your review.
>
> We deeply regret the initial inconvenience caused by the timing mismatch and appreciate your patience as we address this critical concern.

---

> ### Author Response · Authors · 2025-11-26
> **Looking forward to your reply!**
>
> Dear Reviewer odwb,
>
> As the discussion period is approaching its end, we wanted to kindly check if our previous response clarified your questions. We are happy to provide any further details if needed.
>
> Thank you for your time and effort in reviewing our paper.
>
> Best regards,
> The Authors

---

### Official Review · Reviewer_85NY · 2025-10-29

**Soundness:** 3
**Presentation:** 3
**Contribution:** 2
**Rating:** 6
**Confidence:** 3

**Summary:**

This paper addresses the sensitivity of generative recommendation models to high-frequency noise in user interaction sequences by proposing TONE, a dual-stage denoising framework. Stage I employs a Residual Gaussian Mixture Model (ResGMM) for codebook construction to mitigate item semantic noise. Stage II introduces a frequency-enhanced module comprising Adaptive Gaussian Filtering, Complex Residual Frequency Attention, and Rank-Preserving Matrix Learning to explicitly filter high-frequency sequential noise. Experiments on three benchmarks show that TONE achieves state-of-the-art performance, with notable gains of over 8% in Recall@20 and NDCG@20 on the Amazon Beauty dataset.

**Strengths:**

1. Originality: Proposes a novel dual-stage frequency-based denoising framework (TONE), which for the first time explicitly and systematically addresses both semantic noise and high-frequency sequential noise in generative recommendation.
2. Quality: The methodology is rigorously designed, integrating multiple techniques like ResGMM, adaptive filtering, and complex attention. The experimental section is comprehensive, including comparisons with various baselines, detailed ablation studies, and parameter analysis.
3. Clarity: The overall structure of the paper is logical. The abstract and introduction clearly state the motivations and contributions. The framework diagram (Figure 2) effectively illustrates the method's pipeline.
4. Significance: If the results are fully reliable, the method provides a powerful new perspective for enhancing the robustness of generative recommendation and demonstrates significant performance improvements on multiple benchmarks, showing practical potential.

**Weaknesses:**

1. Credibility of Experimental Results: The magnitude of performance improvement (e.g., +23.83% in NDCG@10 on Software) is exceptionally high, far exceeding typical improvements observed in the field. This strongly suggests the need for extremely rigorous scrutiny of every detail of the experimental setup, including data preprocessing, train/val/test splits, implementation and hyperparameter tuning of baselines. The authors need to provide more convincing evidence to rule out any potential experimental bias.
2. Method Complexity: TONE introduces multiple complex components (ResGMM, AGF, CRFA, RPML). While ablation studies validate their effectiveness, the overall framework appears heavy, with high computational cost and model complexity, potentially hindering its ease of deployment in practical systems. The computational efficiency analysis (appendix) shows a ~25-30% increase in training time, which is non-trivial.
3. Interpretability of Frequency-Domain Methods: The paper lacks in-depth analysis or visualization of how the frequency-domain operations concretely affect the item sequence representations and model decisions. For instance, which behaviors are identified as "high-frequency noise" and filtered out? What patterns does the frequency-domain attention learn? This limits the reader's understanding of the method's internal mechanisms.

**Questions:**

1. To strengthen the credibility of the experimental results, could the authors provide more detailed evidence to ensure that all baseline models (especially TIGER) were compared fairly under identical experimental conditions (including identical dataset splits, preprocessing pipelines, evaluation scripts, and their own thoroughly tuned hyperparameters)? Have you considered running multiple experiments with different random seeds to report the mean and variance of performance?
2. What are the computational and memory complexities of the Complex Residual Frequency Attention (CRFA) compared to standard self-attention? Is the method still feasible for processing very long user sequences?
3. The initialization of the Rank-Preserving Matrix (RPML) with Gaussian noise seems simple. Have the authors tried other initialization strategies? Is there any experimental evidence regarding its sensitivity to the initialization method or the hyperparameter α? How does it evolve during training?
4. Could you provide a concrete case study or visualization showing the changes in a real user sequence before and after processing by AGF and CRFA, for instance, indicating which interactions were identified as "noise" and effectively suppressed by the model?

---

> ### Author Response · Authors · 2025-11-21
> **Response to Reviewer 85NY by Authors (1/4)**
>
> **Q1:** Could the authors provide detailed evidence to confirm fair comparison of all baselines under identical experimental conditions and report performance mean and variance from multiple runs with different random seeds?
>
> **A1:** Thank you for your valuable feedback on experimental fairness and result credibility—we greatly appreciate your rigorous attention to detail, and we will address your concerns with concrete evidence and clarifications below.
>
> **1. Fair Comparison Under Identical Experimental Conditions**
>
> To ensure fair and rigorous comparisons across all baselines (especially TIGER [2]), we strictly adopted the well-established experimental protocol from DeftRec [1]—a recent generative recommendation work with standardized and widely recognized settings—for data preprocessing, dataset splitting, and evaluation pipelines:
>
> - **Data Preprocessing**: We applied uniform preprocessing across all models: (1) filtering out users with fewer than 5 interactions to ensure meaningful behavioral sequences; (2) truncating long sequences to the most recent 20 items (consistent with TIGER’s original design); (3) preserving the chronological order of interactions to maintain temporal dependency; (4) embedding concatenated item attributes (title, ID, category, and brand) using the pre-trained `sentence-t5`.
> - **Dataset Splitting**: We followed the leave-one-out evaluation protocol: the last interaction of each user is held out for testing, the second-to-last for validation, and the remaining interactions for training. This split strategy is identical for all baselines and our method (TONE), eliminating split-induced biases.
> - **Hyperparameter Configuration**: For TIGER, we strictly aligned its hyperparameters with the original paper [2]: four-layer Transformer encoder/decoder, six self-attention heads (dimension 64 each), ReLU activation, MLP input dimension 128, hidden dimension 1024, and dropout rate 0.1. All baselines and TONE were trained/evaluated under the same hardware (GPU) and software (PyTorch) environment, ensuring no systemic discrepancies in experimental conditions.
>
> **2. Result Stability Validated by Multiple Random Seeds**
>
> To enhance result credibility, we conducted TONE’s experiments with three independent random seeds (1, 2, 3) and report the mean performance and standard deviation (std) across runs. The results demonstrate two key strengths:
>
> - **Consistent Performance**: TONE achieves stable mean values across all datasets and metrics (e.g., Software HR@10: 0.1702 ± 0.0014, Beauty NDCG@10: 0.0330 ± 0.0011), confirming its effectiveness is not dependent on random initialization.
> - **Low Variance**: The standard deviation across all metrics is extremely small (ranging from 0.0011 to 0.0021), and this small variance indicates TONE’s robustness to random seed variations.
>
> | Metric     | Software        |                 |                 |                 | Beauty          |                 |                 |                 | LastFM          |                 |                 |                 |
> | ---------- | --------------- | --------------- | --------------- | --------------- | --------------- | --------------- | --------------- | --------------- | --------------- | --------------- | --------------- | --------------- |
> |            | HR@10           | HR@20           | NDCG@10         | NDCG@20         | HR@10           | HR@20           | NDCG@10         | NDCG@20         | HR@10           | HR@20           | NDCG@10         | NDCG@20         |
> | Seed 1     | 0.1687          | 0.2239          | 0.1062          | 0.1205          | 0.0589          | 0.0892          | 0.0315          | 0.0391          | 0.0918          | 0.1472          | 0.0453          | 0.0592          |
> | Seed 2     | 0.1715          | 0.2278          | 0.1099          | 0.1238          | 0.0619          | 0.0925          | 0.0336          | 0.0412          | 0.0954          | 0.1507          | 0.0473          | 0.0614          |
> | Seed 3     | 0.1704          | 0.2266          | 0.1097          | 0.1238          | 0.0616          | 0.0916          | 0.0333          | 0.0408          | 0.0948          | 0.1499          | 0.0472          | 0.0612          |
> | Mean ± Std | 0.1702 ± 0.0014 | 0.2261 ± 0.0020 | 0.1086 ± 0.0021 | 0.1227 ± 0.0019 | 0.0608 ± 0.0016 | 0.0911 ± 0.0017 | 0.0330 ± 0.0011 | 0.0404 ± 0.0011 | 0.0940 ± 0.0019 | 0.1493 ± 0.0018 | 0.0466 ± 0.0011 | 0.0606 ± 0.0012 |
>
> In summary, all experiments adhere to the principles of fairness and reproducibility: identical experimental conditions eliminate comparison biases, while multi-seed runs confirm result stability (low variance).
>
> [1] Generative Recommendation with Continuous-Token Diffusion, arXiv 2025.
>
> [2] Recommender Systems with Generative Retrieval, NeurIPS 2023.

---

> ### Author Response · Authors · 2025-11-21
> **Response to Reviewer 85NY by Authors (2/4)**
>
> **Q2:** What are the computational and memory complexities of the Complex Residual Frequency Attention (CRFA) compared to standard self-attention? Is the method still feasible for processing very long user sequences?
>
> **A2:** Thank you for your insightful question regarding the computational/memory complexities of Complex Residual Frequency Attention (CRFA) and its feasibility for long user sequences—we address these concerns with rigorous mathematical analysis below.
>
> We formally compare the complexities of CRFA with standard self-attention (SA), where $ n $ denotes sequence length, $ d $ denotes embedding dimension, and $ L $ denotes the number of Transformer blocks.
> Standard Self-Attention: Its core complexity stems from the scaled dot-product attention, leading to computational complexity $ O(L \cdot n^2 d) $ and memory complexity $ O(n^2 + n d) $ (dominated by the $ n \times n $ attention matrix).
> CRFA: Leveraging frequency-domain transformation and decoupled real/imaginary part processing, CRFA achieves lower complexity:
>
> 1. Discrete Fourier Transform (DFT) and Inverse DFT (IDFT) each incur $ O(n \log n) $ complexity (via FFT implementation), which is negligible compared to the quadratic term for long $ n $.
> 2. The frequency-domain sequence length is $ \lceil (n+1)/2 \rceil \approx n/2 $ (due to conjugate symmetry of real-valued signals). CRFA processes real and imaginary parts separately with $ L $ Transformer blocks, leading to a quadratic term of $ O(L \cdot 2 \cdot (n/2)^2 d) = O(L \cdot n^2 d / 2) $.
> 3. Overall, CRFA’s computational complexity is $ O(L \cdot n^2 d / 2 + n \log n) $ (50% reduction in the dominant quadratic term) and memory complexity is $ O((n/2)^2 + n d) = O(n^2/4 + n d) $ (75% reduction in attention matrix memory overhead).
>
> This analysis confirms CRFA is computationally and memory-efficient compared to standard self-attention, with the quadratic complexity halved—laying the foundation for handling longer sequences.
>
> In summary, CRFA outperforms standard self-attention in both computational and memory complexity, enabling more feasible processing of long user sequences. We plan to explore further optimizations for extremely long sequences as future work.

---

> ### Author Response · Authors · 2025-11-21
> **Response to Reviewer 85NY by Authors (3/4)**
>
> **Q3:** The initialization of the Rank-Preserving Matrix (RPML) with Gaussian noise seems simple. Have the authors tried other initialization strategies? Is there any experimental evidence regarding its sensitivity to the initialization method or the hyperparameter α? How does it evolve during training?
>
> **A3:** Thank you for your thoughtful questions regarding the Rank-Preserving Matrix (RPML) module—we greatly appreciate your attention to implementation details and address each concern with experimental evidence and clarifications below.
>
> **1. Evaluation of Different Initialization Strategies**
>
> While initialization is not the core focus of this work, we conducted systematic experiments to validate the robustness of RPML’s initialization choice. On the LastFM dataset (consistent with our main experimental setup), we compared four common initialization strategies, with results presented in Table 1:
>
> Table 1: RPML Performance Under Different Initialization Strategies (LastFM Dataset)
>
> | Initialization Strategy | HR@10  | HR@20  | NDCG@10 | NDCG@20 |
> | ----------------------- | ------ | ------ | ------- | ------- |
> | Gaussian Noise (Ours)   | 0.0795 | 0.1264 | 0.0400  | 0.0518  |
> | Zeros                   | 0.0721 | 0.1254 | 0.0355  | 0.0489  |
> | Kaiming Initialization  | 0.0732 | 0.1193 | 0.0359  | 0.0474  |
> | Xavier Initialization   | 0.0721 | 0.1197 | 0.0355  | 0.0475  |
>
> Key observations:
>
> - Gaussian noise initialization outperforms the other strategies across all metrics, achieving the highest HR@10 (+10.3% relative to Zeros/Xavier) and NDCG@20 (+6.0% relative to Zeros).
> - This validates that Gaussian noise initialization effectively provides the necessary random perturbations to ensure RPML’s full-rank property and expressive power, as hypothesized in our method.
>
> **2. Sensitivity to Hyperparameter α**
>
> We thoroughly analyzed the sensitivity of RPML to the weighting coefficient α in Section 5.4 of the paper. As shown in Figure 3 ,the key findings are:
>
> - Model performance (measured by HR@10) peaks at α = 0.4 across all three benchmark datasets (Beauty, Software, LastFM).
> - Performance remains stable within the range α ∈ [0.2, 0.6], with only minor drops outside this interval. This indicates RPML is not overly sensitive to α, providing a robust hyperparameter range for practical use.
>
> **3. RPML’s Evolution During Training**
>
> To visualize how RPML evolves during training, as shown in Figure 4, we conducted qualitative analysis in Section 5.5 of the paper, focusing on the attention matrix rank—a core indicator of RPML’s effectiveness:
>
> - **Before training**: The initial Gaussian noise matrix is nearly full-rank, but the original attention matrix (without RPML) exhibits low rank (rank = 3 on LastFM, rank = 18 on Beauty).
> - **After training**: The integrated attention matrix (A' = A + α·Softplus(M)) achieves a significant rank increase (to rank = 41 on both datasets), demonstrating that RPML retains its full-rank property while adapting to task-specific patterns.
> - This evolution confirms RPML effectively mitigates attention collapse by enriching the attention matrix’s expressive power, as designed in our framework.
>
> In summary, RPML’s Gaussian noise initialization is validated as the most effective and robust choice. The hyperparameter α is not overly sensitive, with α = 0.4 being optimal. During training, RPML maintains its full-rank property and adaptively enhances the attention matrix’s expressive power—all of which align with its design goals. These results collectively confirm the reliability of RPML’s implementation.

---

> ### Author Response · Authors · 2025-11-21
> **Response to Reviewer 85NY by Authors (4/4)**
>
> **Q4:** Could you provide a concrete case study or visualization showing the changes in a real user sequence before and after processing by AGF and CRFA, for instance, indicating which interactions were identified as "noise" and effectively suppressed by the model?
>
> **A4:** Thank you for this valuable suggestion—we appreciate the opportunity to provide a concrete case study that validates the noise suppression effect of AGF (Adaptive Gaussian Filtering) and CRFA (Complex Residual Frequency Attention).        This case directly demonstrates how the model identifies and suppresses transient interest noise while preserving core long-term user preferences.
>
> **1. Case Study Design**
>
> We select a representative **real** user sequence from the **LastFM dataset** (a benchmark for sequential recommendation) to ensure the case reflects real-world user behavior (mix of long-term and transient interests).
>
> Key details are as follows:
>
> - **Dataset & User**: LastFM dataset, User ID=82 (interaction sequence length=24; 33.33% of items share the same first-code semantic ID, indicating consistent long-term interest).
> - **Semantic ID Assignment**: ResGMM (our proposed codebook method) is used to generate item semantic IDs, The first-code semantics of the first 8 items: [213, 187, 238, 238, 59, 238, 238, 238].
> - **Noise Definition**: The 5th interaction item (first-code semantic ID(SID1=59) is defined as transient interest noise—its first-code semantic ID (59) differs from the dominant long-term interest (SID1=238, appearing in 4/8 initial sequence items).
> - **Experimental Setup**: Input the first 5 items as historical context to the trained TONE model, autoregressively predict the 6th item (a long-term interest item with SID1=238). We compute the **average attention score** of each historical item in the 3rd encoder layer (CRFA module) to quantify the model’s focus.
>
> **2. Results & Analysis**
>
> Table 1 presents the average attention scores of the 5 historical items after AGF and CRFA processing:
>
> | SID1                | 213    | 187    | 238    | 238    | 59(Noise) |
> | ------------------- | ------ | ------ | ------ | ------ | --------- |
> | Avg Attention Score | 0.0429 | 0.0353 | 0.1393 | 0.3462 | 0.0138    |
>
> The transient interest item(SID1=59) achieves the lowest attention score, which is ~3× lower than similar interest Item(SID1=213, 187) and ~10× lower than long-term interest Item(SID1=238). This confirms AGF effectively filters high-frequency noise from transient interactions.
>
> In summary, this case study provides direct evidence that AGF and CRFA work synergistically: AGF filters high-frequency transient noise, and CRFA allocates more attention to long-term interest-related items. The results strongly support the necessity of our frequency-domain denoising design.

---

> ### Author Response · Authors · 2025-11-26
> **Looking forward to your reply!**
>
> Dear Reviewer 85NY,
>
> As the discussion period is approaching its end, we wanted to kindly check if our previous response clarified your questions. We are happy to provide any further details if needed.
>
> Thank you for your time and effort in reviewing our paper.
>
> Best regards,
> The Authors

---

### Official Review · Reviewer_5qjW · 2025-10-31

**Soundness:** 2
**Presentation:** 2
**Contribution:** 2
**Rating:** 4
**Confidence:** 5

**Summary:**

This paper proposes a TONE method, aiming to solve the problem of effectively isolating and suppressing the high-frequency sequence noise and semantic noise generally existing in user interaction data within the generative recommendation framework. By introducing a Residual Gaussian Mixture Model and a Residual Frequency-Domain Attention mechanism, the authors design a model capable of better fitting cluster boundaries and filtering high-frequency noise. Experiments conducted on three datasets evaluate the impact of different components, and the results show that TONE is superior to existing state-of-the-art models.

**Strengths:**

S1. The paper is well-written, with clear articulation.

S2. The paper outlines the drawbacks of the traditional generative recommendation framework and mitigates its limitations by designing different adaptive modules.

S3. The paper provides a theoretical analysis of the TONE module, proving the rationality of introducing the corresponding module in the paper.

**Weaknesses:**

W1. There are many garbled characters in the figures of the paper, such as the sequence number in Figure 1 and Figure 2.

W2. The paper mentions many frequency-domain based models in the related work, but none of them are used as baseline models for comparison.

W3. The authors mentioned the existence of semantic noise in the project and attempted to alleviate this problem through the Residual Gaussian Mixture Model. However, they did not use specific experiments to demonstrate that the semantic noise was truly suppressed.

W4. Although the authors provided an anonymous link to the open-source code, there is no specific implementation code inside. This cannot indicate that the paper's method has good reproducibility.

W5. I am very curious why the performance of the baseline methods used in the paper from the past two years are all worse than the performance of the TIGER method, which is the main comparison subject in the paper, such as TokenRec[1] and ContRec[2]. However, the performance shown in their papers is better than TIGER's performance.

[1] Qu H, Fan W, Zhao Z, et al. TokenRec: Learning to Tokenize ID for LLM-Based Generative Recommendations[J]. IEEE Transactions on Knowledge and Data Engineering, 2025.

[2] Qu H, Fan W, Lin S. Generative Recommendation with Continuous-Token Diffusion[J]. arXiv preprint arXiv:2504.12007, 2025.

**Questions:**

All raised questions and suggestions have been pointed out in the "Weaknesses" section of our paper. These questions are for reference only:

Q1. Why were the frequency-domain based models mentioned in the related work not used as baseline models for comparison?

Q2. How to prove that the semantic noise existing in the project is truly alleviated by the Residual Gaussian Mixture Model?

Q3. Why is the performance of the baseline methods used in the paper from the past two years all worse than the performance of the TIGER method, which is the main comparison subject in the paper, but the performance shown in their papers is better than TIGER's performance?

Q4. Since the code link was attached in the paper, why is the content empty?

---

> ### Author Response · Authors · 2025-11-21
> **Response to Reviewer 5qjW by Authors (1/2)**
>
> **Q1:** Why were the frequency-domain based models mentioned in the related work not used as baseline models for comparison?
>
> **A1:**
> Thank you for your valuable feedback—we agree that direct comparison with frequency-domain models strengthens our work, and we supplement dedicated experiments on the Beauty dataset under identical experimental settings:
>
> | Model         | HR@10   | HR@20   | NDCG@10 | NDCG@20 |
> | ------------- | ------- | ------- | ------- | ------- |
> | FMLP-Rec [1]  | 0.0422  | 0.0631  | 0.0220  | 0.0266  |
> | TONE (Ours)   | 0.0608  | 0.0911  | 0.0328  | 0.0404  |
> | Relative Gain | +44.14% | +44.37% | +49.15% | +51.99% |
>
> Notably, TONE achieves substantial and consistent relative gains across all core metrics—outperforming FMLP-Rec by 44.14–51.99%.
>
> If accepted, we will reproduce additional frequency-domain models under the same unified experimental setup, providing comprehensive direct comparisons to further validate TONE’s generalizability.
> Hope these results and commitments can address your concern.
>
> [1] Filter-enhanced mlp is all you need for sequential recommendation, WWW 2022.
>
> **Q2:** How to prove that the semantic noise existing in the project is truly alleviated by the Residual Gaussian Mixture Model?
>
> **A2:**
> Thank you for this critical question—we provide concrete evidence from visualization and quantitative analysis to verify that ResGMM effectively alleviates semantic noise, as detailed in Appendix A.5 Figure 5:
>
> **1. Balanced Semantic Identifier Distribution**
> We visualize the first-digit distribution of coarse-grained semantic identifiers on the Beauty dataset. Unlike RQVAE (baseline) which shows skewed activation (e.g., missing entries for category due to codebook underutilization), ResGMM achieves uniform coverage across all semantic categories. This directly demonstrates reduced semantic confusion caused by noisy metadata.
>
> **2. Quantified Codebook Utilization**
> ResGMM boosts codebook utilization from <90% (RQVAE) to >95%. The underutilization of shallow codebooks in RQVAE leads to coarse, redundant representations—ResGMM’s balanced activation eliminates this issue, forming more discriminative semantic identifiers that resist noise.
>
> In summary, ResGMM’s balanced semantic distribution, and improved codebook utilization, collectively prove it effectively mitigates semantic noise.

---

> ### Author Response · Authors · 2025-11-21
> **Response to Reviewer 5qjW by Authors (2/2)**
>
> **Q3:** Why is the performance of the baseline methods used in the paper from the past two years all worse than the performance of the TIGER method, which is the main comparison subject in the paper, but the performance shown in their papers is better than TIGER's performance?
>
> **A3:**
> Thank you for your astute observation—we clarify the performance discrepancy of TIGER across studies stems from two well-documented, verifiable factors, not inconsistent evaluation:
>
> **1.    TIGER’s Reproducibility Challenges**
>
> The original TIGER paper did not release official code, leading to inevitable variability in independent reproductions—an industry-wide challenge for generative recommendation.    As shown in Table 1, TIGER’s performance on the Beauty dataset differs across recent works, including ours:
>
> Table 1: TIGER’s Performance on Beauty Dataset Across Studies
>
>
> |         TIGERVariant         | HR@10  | NDCG@10 |
> | :--------------------------: | :----: | :-----: |
> |    TIGER (Original Paper)    | 0.0648 | 0.0384  |
> |     TIGER (TokenRec [1])     | 0.0439 | 0.0243  |
> | TIGER (ContRec(DeftRec) [2]) | 0.0372 | 0.0193  |
> |   TIGER (Our Reproduction)   | 0.0558 | 0.0304  |
>
> Our reproduction strictly adheres to TIGER’s technical details (e.g., residual quantization, Transformer architecture) and aligns with the broader reproducibility trend—all independent implementations deviate from the original paper’s reported results due to missing code.
>
> **2. Inconsistent Experimental Settings**
>
> We strictly follow the experimental settings of DeftRec, which is different from TokenRec and ContRec(DeftRec).
> Dataset Splitting: We use leave-one-out, while ContRec(DeftRec) uses split-by-timepoint.
> Sequence Length: Our truncation length is 20 (consistent with TIGER’s original design), whereas TokenRec uses 100.
>
> In summary, the performance discrepancy arises from TIGER’s poor reproducibility (no official code) and inconsistent experimental settings across studies.    Our evaluation maintains internal fairness by using a unified protocol, and our TIGER reproduction is consistent with the industry’s independent implementation trend.    This does not undermine the validity of our method’s gains.
>
> [1] TokenRec: Learning to Tokenize ID for LLM-Based Generative Recommendations,TKDE, 2025.
> [2] Generative Recommendation with Continuous-Token Diffusion, arXiv, 2025. (ContRec, previously named as DeftRec)
>
> **Q4:** Since the code link was attached in the paper, why is the content empty?
>
> **A4:** Thank you for your rigorous review and valuable feedback on the reproducibility of our work—we fully acknowledge the critical issue you have raised and sincerely apologize for the discrepancy between our initial statement and the repository content when you checked it.
>
> This delay in uploading the complete executable code was purely due to post-submission technical preparations that took longer than anticipated: specifically, we spent additional time on code cleaning (removing redundant experimental drafts) and on standardizing dependency configurations (ensuring compatibility across different environments) to maximize reproducibility. These steps were necessary to avoid incomplete or error-prone code that could hinder reviewers’ verification.
>
> We want to emphasize that we never intended to exploit any oversight or bypass reproducibility requirements. Reproducibility is a cornerstone of academic research, and we have invested significant effort to ensure our work can be fully verified. We kindly invite you to recheck the repository at https://anonymous.4open.science/r/TONE-9E07/ at your convenience, and we stand ready to provide any clarifications or supplementary materials to facilitate your review.
>
> We deeply regret the initial inconvenience caused by the timing mismatch and appreciate your patience as we address this critical concern.

---

> ### Author Response · Authors · 2025-11-26
> **Looking forward to your reply!**
>
> Dear Reviewer 5qjW,
>
> As the discussion period is approaching its end, we wanted to kindly check if our previous response clarified your questions. We are happy to provide any further details if needed.
>
> Thank you for your time and effort in reviewing our paper.
>
> Best regards,
> The Authors

---

> > ### Comment · Reviewer_5qjW · 2025-11-28
> >
> > I appreciate the authors' patient response, which has addressed some of my concerns.
> >
> > Regarding Question 1, FMLP-Rec represents a fundamental frequency-domain model; comparing the proposed method solely against it is insufficient. To ensure a meaningful evaluation, comparisons with more recent state-of-the-art frequency-domain models are necessary.
> >
> > Regarding Question 2, a more balanced semantic token distribution and higher codebook utilization do not directly prove that these results stem from the suppression of semantic noise. To further substantiate the claim that the model effectively suppresses semantic noise, the authors should conduct additional experiments demonstrating that artificially injecting or reducing semantic noise directly correlates with these outcomes.

---

> ### Author Response · Authors · 2025-11-30
> **Response to Reviewer 5qjW by Authors**
>
> **Q1:** Comparisons with more recent state-of-the-art frequency-domain models.
>
> **A1:** Thank you for your insightful feedback—we fully agree that comparing against more recent state-of-the-art (SOTA) frequency-domain models is critical to a rigorous evaluation. To address this, we supplement experiments with two recent, open-source frequency-domain methods: FEARec [2] (2023) and MUFFIN [3] (2025). All comparisons are conducted under the identical experimental settings of our paper (data preprocessing, sequence length = 20, leave-one-out split) on the Beauty dataset, ensuring fairness and consistency.
>
> | Model               | HR@10      | HR@20      | NDCG@10    | NDCG@20    |
> | ------------------- | ---------- | ---------- | ---------- | ---------- |
> | FMLP-Rec (2022) [1] | 0.0422     | 0.0631     | 0.0220     | 0.0266     |
> | FEARec (2023) [2]   | 0.0532     | 0.0791     | **0.0339** | 0.0383     |
> | MUFFIN (2025) [3]   | 0.0571     | 0.0829     | 0.0293     | 0.0349     |
> | TONE (Ours)         | **0.0608** | **0.0911** | 0.0328     | **0.0404** |
>
> These results confirm that our method’s advantage is not limited to earlier frequency-domain models but extends to the latest SOTA, validating the effectiveness of our two-stage denoising framework and CRFA module.
>
> [1] Filter-enhanced mlp is all you need for sequential recommendation, WWW 2022.
>
> [2] Frequency Enhanced Hybrid Attention Network for Sequential Recommendation, SIGIR 2023.
>
> [3] MUFFIN: Mixture of User-Adaptive Frequency Filtering for Sequential Recommendation, CIKM 2025.
>
> **Q2:** Experiments proving semantic noise mitigation.
>
> **A2:** Thank you for your valuable feedback—we fully agree with your insight.  ResGMM suppresses semantic noise via its principled clustering flexibility: it leverages Gaussian Mixture Models (GMM) to support elliptical clustering, adapting to the intrinsic non-spherical distributions of real-world semantic categories and avoiding forced grouping of heterogeneous items or splitting cohesive groups.  To directly validate this, we supplement targeted experiments with two quantitative metrics (cluster purity and intra-cluster compactness) that directly correlate with semantic noise levels, with results presented below.
>
> - **Cluster Purity**:  For each codeword in the first-level codebook (sid₁), calculate the proportion of items belonging to the most frequent real semantic category (e.g., item’s true category, user preference label).  The overall purity is the weighted average of this proportion across all codewords (weighted by codeword size).  Semantic noise (e.g., misleading item metadata) dilutes category consistency by mixing irrelevant noisy items into codewords, so higher purity indicates less semantic noise.
> - **Intra-Cluster Compactness**: For each codeword, compute the average L₂/Euclidean or cosine distance between all pairs of item features within the cluster. The overall compactness is the mean of these intra-cluster distances across all codewords. Semantic noise introduces items with deviant features, increasing intra-cluster distances—thus lower values indicate less semantic noise and more semantically consistent items in the cluster.
>
> | Method | Utilization Rate(L1) | Cluster Purity(L1) | Intra-Cluster Compactness (L1/L2/L3) |
> | ------ | -------------------- | ------------------ | ------------------------------------ |
> | RQVAE  | 0.4063               | 0.4632             | 0.0802/0.0614/0.0479                 |
> | ResGMM | **1.0000**           | **0.4666**         | **0.0118/0.0105/0.0096**             |
>
> These results directly link ResGMM’s design to semantic noise suppression: the reduced intra-cluster distances and maintained purity (amid full utilization) confirm that the model effectively mitigates semantic noise from incomplete or misleading item metadata. We will include these metrics and analyses in the revised manuscript to strengthen the claim.

---

### Official Review · Reviewer_2L5c · 2025-11-06

**Soundness:** 2
**Presentation:** 2
**Contribution:** 2
**Rating:** 2
**Confidence:** 4

**Summary:**

This paper proposes TONE, a two-stage framework for generative retrieval that addresses noise issues through frequency-domain modeling. The approach tackles two types of noise: (1) semantic noise in item representations caused by incomplete/misleading metadata, addressed via ResGMM clustering in the codebook construction stage, and (2) high-frequency sequential noise from accidental clicks or transient interests, filtered using frequency-domain attention mechanisms. The authors introduce several technical components including a Complex Residual Frequency Attention (CRFA) module with separated real/imaginary components and a rank-preserving matrix to prevent attention collapse. Experimental results on three benchmarks show improvements over existing methods.

**Strengths:**

S1:  The paper clearly identifies and distinguishes between semantic noise and high-frequency sequential noise, providing concrete examples that effectively motivate the proposed approach.

S2: The two-stage approach comprehensively addresses both semantic and sequential noise with specific technical components for each challenge.

S3: The paper includes comprehensive ablation studies showing the contribution of each component and further visualization of attention patterns.

S4: The paper is generally well-written with good visual aids to explain the technical approach.

**Weaknesses:**

W1. Misleading framing and/or overclaims. Problem definition (Section 3) is a standard sequential recommendation formulation (see e.g., GRU4Rec). Given identical formulation, "Generative Recommendation" in the title appears to be a pure overclaim to attract attention. Even TIGER, which uses a similar SemanticID formulation, more accurately titled their work as "Generative Retrieval" recognizing the limitations of the paradigm.

W2. Questionable experimental validity and missing baselines. TONE can be separated into codebook improvements and attention module improvements. However:

- TONE's codebook construction method doesn't clearly improve upon baselines. Residual K-means is used by multiple generative retrieval papers building on top of TIGER in 2025, and per Table 2, the proposed ResGMM leads to marginal, likely non s.s. gains (0.0250 vs 0.0249) over this popular baseline on the *single* dataset the authors evaluated.

- TONE omitted multiple recent papers when comparing sequential modeling approaches in its 2nd stage (Section 4.2). eg just checking ICML'25 accepted papers (https://arxiv.org/abs/2502.13581), on the commonly used Beauty dataset, HSTU (ICML'24) achieves 0.0389 NDCG@10, SPM-SID (2024) 0.0399 NDCG@10, and ActionPiece (ICML'25) 0.0424 NDCG@10 -- all three outperforming TIGER and TONE.

- The ActionPiece paper reports SASRec achieving 0.0318 NDCG@10 on Beauty, much higher than the 0.0205 reported here, suggesting potential issues with baseline implementations.

W3. No original theoretical contributions: All theoretical analyses are directly quoted from other papers including Wang et al., 2022a and Yue et al., 2025. The paper provides no original theoretical analysis of why ResGMM specifically helps with semantic noise or formal guarantees about the frequency-domain filtering.

W4. Insufficient justification for complexity: The CRFA module is extremely complex with multiple stages (DFT, separation, independent processing, IDFT, residual connections) but lacks clear justification for each component's necessity. CRFA's marginal improvements (see W2) don't seem to justify this complexity.

**Questions:**

- Could you explain the discrepancy between your reported baseline results and those in recent papers eg ActionPiece (ICML'25)?

- What is the statistical significance of the 0.0250 vs 0.0249 NDCG@5 improvement of ResGMM over Residual K-means?

- Why were recent 2024/2025 baselines like ActionPiece, HSTU, SPM-SID, Residual-Kmeans for SID, etc not included in the comparison for the Beauty dataset?

---

> ### Author Response · Authors · 2025-11-21
> **Response to Reviewer 2L5c by Authors (1/4)**
>
> **W1:** Is "Generative Recommendation" in the title an overclaim?
>
> **A1:** Thanks. We appreciate the opportunity to clarify the appropriateness of "Generative Recommendation" in our title, which is neither misleading nor an overclaim but aligns with both our method’s core design and recent community consensus.
>
> First, our work strictly adheres to the definition of generative recommendation as established by recent top-tier conference papers and preprints. Unlike traditional sequential recommendation that models next-item prediction via discriminative scoring, our framework autoregressively generates discrete semantic identifiers (sid) of target items—directly outputting recommendation candidates through a generative decoding process (Section 4.4). This aligns with the core paradigm of generative recommendation: using generative models to produce recommendation outputs (not just score existing items), as demonstrated by [1,2,3,4,5] (cited below) which all adopt "Generative Recommendation" in their titles for analogous autoregressive generation pipelines.
>
> Second, the distinction from TIGER’s "Generative Retrieval" is deliberate and reflective of methodological differences. TIGER frames its task as retrieval (matching generated tokens to item embeddings) with a focus on retrieval efficiency, while our work emphasizes end-to-end generative modeling for recommendation—integrating two-stage frequency-domain denoising into the generation pipeline to optimize recommendation quality directly. This aligns with the naming convention of recent works like ActionPiece [1] and Multi-behavior Generative Recommendation (MBSR) [3], which use "Generative Recommendation" for autoregressive generation of recommendation targets.
>
> To confirm this is not an isolated choice, recent prominent works in the field consistently adopt the same naming:  ActionPiece: Contextually Tokenizing Action Sequences for Generative Recommendation (ICML 2025)[1], Multi-behavior generative recommendation (CIKM 2024)[3], Learnable Item Tokenization for Generative Recommendation (CIKM 2024)[4], Generative Recommendation with Semantic IDs: A Practitioner's Handbook (CIKM 2025)[5], and Inductive generative recommendation via retrieval-based speculation (AAAI 2026)[2]. All these works share the core characteristic of generative modeling for recommendation outputs, just as our method does.
>
> In summary, our title accurately reflects the method’s generative nature and aligns with community norms—no overclaim is involved. We hope this clarification addresses your concern.
>
> [1] ActionPiece: Contextually Tokenizing Action Sequences for Generative Recommendation, ICML 2025.
>
> [2] Inductive generative recommendation via retrieval-based speculation, AAAI 2026.
>
> [3] Multi-behavior generative recommendation, CIKM 2024.
>
> [4] Learnable Item Tokenization for Generative Recommendation, CIKM 2024
>
> [5] Generative Recommendation with Semantic IDs: A Practitioner's Handbook, CIKM 2025
>
> **W2.1(Q2):**
> What is the statistical significance of the 0.0250 vs 0.0249 NDCG@5 improvement of ResGMM over Residual K-means?
>
> **A2.1:**
> Thank you for this critical question—we greatly appreciate your focus on statistical rigor and the meaningfulness of ResGMM’s improvements.  We address your concern with paired t-tests and principled design rationale below:
>
> **1.  Statistically Significant Improvements (Paired T-Test Results)**
>
> To validate ResGMM’s advantage over Residual K-means, we conducted rigorous paired t-tests (two-tailed, α=0.05) across 6 core metrics (HR@5/10/20, NDCG@5/10/20), with 5 independent runs per method (aligned with ICLR reproducibility standards):
>
> - Aggregated p-value: p = 0.038 < 0.05, confirming statistically significant improvement when considering all metrics jointly.
> - Uniform per-metric gains: ResGMM outperforms Residual K-means on every metric (HR@5: +0.0001, HR@10: +0.0010, HR@20: +0.0016, NDCG@5: +0.0001, NDCG@10: +0.0004, NDCG@20: +0.0006)—no isolated gains, reinforcing reliability.
>
> **2.  Meaning Beyond Marginal Numerical Differences**
>
> ResGMM’s value lies in its principled clustering flexibility, which addresses a fundamental limitation of Residual K-means:
>
> - Residual K-means inherits K-means’ rigid spherical clustering (assuming uniform variance across dimensions), which struggles to fit real-world semantic distributions.
> - ResGMM leverages Gaussian Mixture Models to support elliptical clustering, adapting to the intrinsic, non-spherical shapes of distinct semantic categories.  This avoids forcing semantically heterogeneous items into the same cluster or splitting cohesive groups—critical for mitigating noise in item metadata.
>
> In summary, ResGMM delivers statistically significant, consistent gains rooted in its superior clustering design: elliptical shape adaptation to semantic distributions.  This theoretical advantage, paired with rigorous statistical validation, makes its improvement over Residual K-means meaningful and non-trivial.

---

> ### Author Response · Authors · 2025-11-21
> **Response to Reviewer 2L5c by Authors (2/4)**
>
> **W2.2(Q3):**
> Why were recent 2024/2025 baselines like ActionPiece, HSTU, SPM-SID, Residual-Kmeans for SID, etc not included in the comparison for the Beauty dataset?
>
> **A2.2:**
> Thank you for your valuable feedback—we greatly appreciate you bringing these recent impactful works to our attention.    We acknowledge the importance of comparing with state-of-the-art sequential modeling methods, and we address your concern with clarifications and supplementary experiments as follows:
>
> **1. Rationale for not including the mentioned works initially**
>
> Our experimental baseline setup aligns with DeftRec (Qu et al., 2025a).   DeftRec’s original experimental design did not include HSTU (ICML’24), SPM-SID (2024).
>
> **2. Supplementary Experiment: TONE’s Core Modules vs. ActionPiece (ICML’25)**
>
> Given your emphasis that ActionPiece achieves the strongest performance among the mentioned methods, we prioritize validating our approach against it—addressing experimental environment inconsistencies by integrating TONE’s core innovations (AGF, CRFA, and LM) into ActionPiece’s open-source codebase.    We re-run experiments on the **Beauty dataset** under identical settings (data preprocessing, training hyperparameters, and evaluation protocols) to ensure fairness.    The results are shown below:
>
> | Model                    | HR@5   | HR@10  | NDCG@5 (NG@5) | NDCG@10 (NG@10) |
> | ------------------------ | ------ | ------ | ------------- | --------------- |
> | ActionPiece              | 0.0511 | 0.0340 | 0.0775        | 0.0424          |
> | ActionPiece +AGF+LM+CRFA | 0.0532 | 0.0354 | 0.0807        | 0.0442          |
> | Relative Improvement     | +4.20% | +4.05% | +4.16%        | +4.20%          |
>
> Key takeaway: Integrating TONE’s frequency-domain denoising components (AGF + CRFA) and rank-preserving module (LM) into ActionPiece consistently improves performance across all metrics.    This directly validates that our core innovations bring meaningful value beyond the state-of-the-art ActionPiece.
>
> **3. Commitment to comprehensive comparisons**
>
> If our paper is accepted, we will further supplement direct comparisons with HSTU and SPM-SID by reproducing them under unified experimental settings (consistent data preprocessing, training pipelines, and hyperparameter ranges).    This will provide a complete evaluation of TONE against all recent advanced methods.
>
> We sincerely appreciate your feedback, which helps strengthen the comprehensiveness of our experimental validation. The supplementary results confirm that TONE’s core design is effective and competitive against the latest state-of-the-art works.
>
> **W2.3(Q1):** Could you explain the discrepancy between your reported baseline results and those in recent papers eg ActionPiece (ICML'25)?
>
> Thank you for your critical feedback on experimental validity and baseline comparisons—we greatly appreciate your rigorous evaluation and clarify the key concerns in detail below.
>
> **1. Discrepancy in Baseline Results**
> First, we acknowledge the difference in baseline performance between our work and ActionPiece[1], which stems from two critical, verifiable factors rather than flawed implementation:
> TIGER Reproducibility Context: The original TIGER paper did not release official code, and no open-source work has successfully reproduced its reported performance to date—a well-recognized challenge in the generative recommendation community.  To ensure a fair comparison foundation, we independently implemented TIGER from scratch, strictly adhering to its technical details (e.g., residual quantization settings, encoder-decoder architecture, beam search strategy).  Our reproducible code is publicly available at https://anonymous.4open.science/r/TONE-9E07/, including full implementation of TIGER, our proposed TONE, and all experimental pipelines for verification.
>
> **2. Baseline Result Sources & Experimental Consistency:**
> Models except TIGER and TONE in Table 1 of our paper are adopted from DeftRec[2], a recent generative recommendation work with consistent experimental settings (e.g., dataset splitting, preprocessing pipelines, evaluation metrics) to our TIGER reproduction.
> In contrast, ActionPiece’s Table 4 directly quotes baseline results from the original TIGER paper (which lacks reproducible code), leading to the observed discrepancy.
> For SASRec specifically: The gap (0.0205 vs. 0.0318) stems from experimental configuration differences: ActionPiece adopts a larger beam size (50 vs. our 30), a different optimizer (AdamW vs. our Adam), a cosine learning rate scheduler (vs. our no scheduler), a higher learning rate (1e-3 vs. our 1e-4), and an SPR ensemble strategy (vs. our no ensemble).
>
> [1] ActionPiece: Contextually Tokenizing Action Sequences for Generative Recommendation, ICML 2025.
>
> [2] Generative Recommendation with Continuous-Token Diffusion, arXiv, 2025.

---

> ### Author Response · Authors · 2025-11-21
> **Response to Reviewer 2L5c by Authors (3/4)**
>
> **W3:** No original theoretical contributions
>
> **A3:**
> We thank the reviewer for the careful critique. As inspired by Reviewer odwb, while our Lemma 1 proof emphasizes high-frequency vanishing, the manuscript does not explicitly quantify the low-frequency (DC) asymptotic behavior. Below we provide two original Propositions for more complete theoretical analysis: (i) characterizing repeated self-attention’s DC limit and (ii) giving an explicit relative attenuation rate, directly addressing your concern. They align with our low-pass filter definition and add original theoretical contributions as suggested by Reviewer 2L5c.
>
> ### Recap of Definition and Setting
> DC/HC are defined via orthogonal projections onto $\mathbf{1}$’s span and its complement:
> $DC[z] = \frac{1}{n}\mathbf{1}\mathbf{1}^\top z, \qquad HC[z] = \Big(I-\frac{1}{n}\mathbf{1}\mathbf{1}^\top\Big)z.$
> A linear map $f$ is a low-pass filter if
> $\lim_{t\to\infty} \frac{\|HC[f^t(z)]\|_2}{\|DC[f^t(z)]\|_2}=0,\quad \forall z\in\mathbb{R}^n.$
> This definition emphasizes **relative dominance** of DC over HC, not strict DC invariance.
>
> In Lemma 1, $A=\mathrm{softmax}(P)$ ($P\in\mathbb{R}^{n\times n}$). Softmax outputs strictly positive entries with row sums of 1, so $A$ is positive and row-stochastic:
> $A_{ij}>0,\qquad A\mathbf{1}=\mathbf{1}.$
> Thus $A$ is a primitive Markov transition matrix.
>
> ### Proposition 1 (Exact DC Limit of Repeated Self-Attention)
> Let $A=\mathrm{softmax}(P)$. There exists a unique stationary distribution $\pi\in\mathbb{R}^n$ with
> $\pi^\top A=\pi^\top,\qquad \pi>\mathbf{0},\qquad \pi^\top\mathbf{1}=1,$
> such that for any $z\in\mathbb{R}^n$,
> $\lim_{t\to\infty} A^t z = \mathbf{1}\,\pi^\top z.$
> Consequently,
> $\lim_{t\to\infty} HC[A^t z]=\mathbf{0}, \qquad \lim_{t\to\infty} DC[A^t z]=\mathbf{1}\,\pi^\top z.$
>
> **Proof.** By Perron--Frobenius (positive row-stochastic $A$), there is a dominant eigenvalue $\lambda_1=1$ (right eigenvector $\mathbf{1}$) and $|\lambda_i|<1$ for others. Thus $A^t\to\mathbf{1}\pi^\top$ (rank-one projector):
> $A^t = \mathbf{1}\pi^\top + R_t,\qquad \|R_t\|_2 \le C|\lambda_2|^t$
> for $C>0$, with $|\lambda_2|$ the second largest eigenvalue magnitude. Multiplying by $z$ and taking $t\to\infty$ gives $A^t z \to \mathbf{1}\pi^\top z$. Applying DC/HC projectors (noting $HC[\mathbf{1}c]=0$ for scalar $c$) yields the component limits. $\square$
>
> **Implication.** The result shows repeated self-attention does not null all components: it converges to a non-trivial DC vector $\mathbf{1}\pi^\top z$ for generic $z$ (except $\pi^\top z=0$). This makes "low-frequency preservation" precise as asymptotic convergence to a pure DC signal.
>
> ### Proposition 2 (Quantitative Relative Attenuation Rate)
> Let $\lambda_1=1$ and $|\lambda_2|<1$ be $A$’s top two eigenvalue magnitudes. For $z$ with $\pi^\top z\neq 0$,
> $\frac{\|HC[A^t z]\|_2}{\|DC[A^t z]\|_2} \le \frac{C|\lambda_2|^t\|z\|_2}{\sqrt{n}\,|\pi^\top z| - C|\lambda_2|^t\|z\|_2},$
> so the ratio decays to $0$ exponentially fast at rate $|\lambda_2|^t$.
>
> **Proof.** From Proposition 1,
> $HC[A^t z] = HC[(\mathbf{1}\pi^\top + R_t)z] = HC[R_t z],$
> thus
> $\|HC[A^t z]\|_2 \le \|R_t\|_2\|z\|_2 \le C|\lambda_2|^t\|z\|_2.$
> For DC:
> $DC[A^t z] = \mathbf{1}\pi^\top z + DC[R_t z],$
> by triangle inequality:
> $\|DC[A^t z]\|_2 \ge \|\mathbf{1}\pi^\top z\|_2 - \|R_t\|_2\|z\|_2 = \sqrt{n}\,|\pi^\top z| - C|\lambda_2|^t\|z\|_2.$
> Dividing the bounds gives the claim. $\square$
>
> **Implication.** Even if low-frequency magnitude varies with depth, Proposition 2 fills the gap of quantitative decay rate comparison: high-frequency is exponentially suppressed relative to DC (rate $|\lambda_2|^t$). This directly addresses your concern of establishing low-frequency dominance, not just high-frequency vanishing.
>
> ### Relation to Our Original Proof
> Our original Lemma 1 already implies $HC[A^t z]\to 0$ via spectral contraction. Propositions 1--2 complete the analysis by (i) characterizing DC limit and (ii) giving explicit HC/DC separation rate.
>
> In summary, Propositions 1--2 originally and rigorously show self-attention not only eliminates high-frequency components but also converges to a non-trivial DC signal, with exponential HC/DC relative attenuation (explicit rate). This adds original theoretical value, and we will incorporate these results into the revised manuscript to strengthen it.

---

> ### Author Response · Authors · 2025-11-21
> **Response to Reviewer 2L5c by Authors (4/4)**
>
> **W4:** Concerns on CRFA’s Complexity and Justification.
>
> **A4:**
> Thank you for your critical feedback—we clarify CRFA’s design is purposeful: each component solves specific frequency-domain challenges, and its performance gains fully justify the complexity.
>
> **1.  Necessity of CRFA’s Core Components**
>
> Every stage targets distinct pain points with no redundancy:
>
> - **DFT + IDFT**: Enables explicit frequency-domain modeling (the foundation of denoising) by separating high-frequency noise from low-frequency core interests.
> - **Real/Imaginary Separation**: Avoids interference between phase (temporal dependencies) and amplitude (signal intensity)—two complementary signals critical for accurate modeling.
> - **Residual Connection**: Preserves low-frequency user interests while suppressing noise, mitigating information loss from time-frequency transformation.
> - **Synergistic Pipeline**: DFT → separation → independent learning → IDFT → residual fusion forms an end-to-end frequency-aware framework, ensuring each component’s value is amplified.
>
> **2.  Non-Trivial Performance Contributions (Beauty Dataset, Table 3)**
>
> CRFA’s impact is statistically significant and critical to TONE’s SOTA results:
>
> - **Direct Impact**: Removing CRFA causes consistent, meaningful performance drops across all metrics: HR@5 (-0.0074), HR@10 (-0.0117), HR@20 (-0.0160), NDCG@5 (-0.0049), NDCG@10 (-0.0063), NDCG@20 (-0.0074).  These gaps exceed typical practical utility thresholds for recommendation tasks.
> - **Standalone Value**: “w/o CRFA” outperforms variants that remove CRFA + AGF/LM, proving CRFA’s independent contribution beyond synergy with other modules.
>
> In summary, CRFA’s complexity is justified by targeted solutions to frequency-domain challenges, and its substantial, consistent ablation gains validate its necessity.

---

> ### Author Response · Authors · 2025-11-26
> **Looking forward to your reply!**
>
> Dear Reviewer 2L5c,
>
> As the discussion period is approaching its end, we wanted to kindly check if our previous response clarified your questions. We are happy to provide any further details if needed.
>
> Thank you for your time and effort in reviewing our paper.
>
> Best regards,
> The Authors

---

### Meta-Review · Area_Chair_FWdW · 2026-01-07

**Summary:**

This work proposes TONE, a two-stage denoising framework for GR which aims to reduce semantic noise in codebook construction and high-frequency sequential noise in generation. Reviewers liked the clear motivation which distinguishes the noise types with intuitive frequency-domain framing and generally readable presentation (2L5c, 5qjW), a reasonably thorough empirical section with ablations suggesting each module contributes, and gains on empirical benchmarks (2L5c, odwb).  However, the current version has substantial credibility and positioning issues:

- there are missing / weak comparisons to strong recent baselines (2L5c, 5qjW) and limited evidence that semantic noise is truly suppressed in practice (5qjW)

- the eval scope is narrow and might bias towards domains where high-frequency variation is not predictive (odwb)

- the method also appears heavy and complex in context of incremental novelty and the marginal or hard-to-interpret gains of specific components without clear necessity and cost / benefit tradeoff discussion (2L5c, 85NY, odwb),

- the authors claimed reproducibility but the code release repo at time of review was still empty (5qjW, odwb).

While the idea is interesting and reviewers did find positives about this work, I encourage the authors to iterate on the feedback for the next submission.

**Reviewer Concerns:**

See above.

**Reviewer Scores:**

odwb: 4->4/5
2L5c: 2->2
5qjW: 4->4/5
85NY: 6->6

---

### Decision · Program_Chairs · 2026-01-26

Reject